**Foreign and domestic contributions to springtime ozone over China**
Ruijing Ni[1], Jintai Lin[1], Yingying Yan[1], Weili Lin[2]
[1]Laboratory for Climate and Ocean-Atmosphere Studies, Department of Atmospheric
and Oceanic Sciences, School of Physics, Peking University, Beijing 100871, China
[2]College of Life and Environmental Sciences, Minzu University of China, Beijing
100081, China
*Correspondence to:* J.-T. Lin (linjt@pku.edu.cn)
**Abstract.** China is facing a severe ozone problem, but the origin of its ozone remains
unclear. Here we use a GEOS-Chem based global-regional two-way coupled model
system to quantify the individual contributions of eight emission source regions
worldwide to springtime ozone in 2008 over China. The model reproduces the observed
ozone from 31 ground sites and various aircraft and ozonesonde measurements in China
and nearby countries, with a mean bias of 10–15% both near the surface and in the
troposphere. We then combine zero-out simulations, tagged ozone simulations, and a
linear weighting approach to accounting for the effect of nonlinear chemistry on ozone
source attribution. We find considerable contributions of total foreign anthropogenic
emissions to surface ozone over China (2–11 ppb). For ozone averaged over China of
anthropogenic origin, foreign regions together contribute 40–50% below the height of
2 km and 85% in the upper troposphere. For total foreign anthropogenic emissions
contributed ozone over China at various heights, the portion of transboundary ozone
produced within foreign emission source regions is less than 50%, with the rest
produced by precursors transported out of those source regions. Japan and Korea
contribute 0.6–2.1 ppb of surface ozone over the east coastal regions. South-East Asia
contributes 1–5 ppb over much of southern China and South Asia contributes up to 5–
10 ppb of surface ozone over border of southwestern China; and their contributions
increase with height due to strong upwelling over the source regions. European
contribution reaches 2.1–3.0 ppb for surface ozone over the northern border of China
and 1.5 ppb in the lower troposphere averaged over China. North America contributes
0.9–2.7 ppb of surface ozone over most of China (1.5–2.1 ppb over the North China
Plain), with a China average at 1.5–2.5 ppb at different heights below 8 km, due to its
large anthropogenic emissions and the transport-favorable mid-latitude westerly. In
addition to domestic emission control, global emission reduction is critical for China's
ozone mitigation.

## 1. Introduction

Ozone is an important atmospheric oxidant and the primary source of the hydroxyl radical (OH). At the surface, ozone also damages human health and reduces crop yield. China is currently facing a severe ozone pollution problem, with measured maximum hourly ozone exceeding 200 ppb in many cities (Wang et al., 2006; Xue et al., 2014). Even in the remote areas of western China, measured daily mean concentrations of ozone exceed 50 ppb frequently (Xue et al., 2011; Lin et al., 2015). Xu et al. (2016) showed that daytime ozone at Waliguan, a global background station, grew significantly from 1994 to 2013 at a rate of $0.24\pm0.16$ ppb year$^{-1}$. The severe ozone problem is largely associated with growth in anthropogenic emissions of nitrogen oxides (NOx) and non-methane volatile organic compounds (NMVOC). Chinese anthropogenic NOx emissions increased at a rate of 7.9% year$^{-1}$ from 2000 to 2010 (Zhao et al., 2013); and its anthropogenic NMVOC emissions increased from 22.45 Tg in 2008 to 29.85 Tg in 2012 (Wu et al., 2016).

Ozone has a lifetime of several days to weeks in the troposphere (Young et al., 2013;Yan et al., 2016), which makes its long-distance transport across regions and even continents possible. Many observational and modeling studies have showed substantial trans-Pacific and trans-Atlantic transport of ozone and precursors (Jacob et al., 1999; Derwent et al., 2004; Lin et al., 2008; Cooper et al., 2010; Verstraeten et al., 2016). The trans-Pacific transport of East Asian air pollutants enhances springtime surface ozone concentrations over the western United States by 1–5 ppb (Zhang et al., 2008; Brown-Steiner and Hess, 2011; Lin et al., 2012b; Lin et al., 2014). Auvray and Bey (2005) reported that North American and Asian ozone account for 10.9% and 7.7% of ozone over Europe, respectively. The Hemispheric Transport of Air Pollution (HTAP) project studied the trans-continental pollution, by model sensitivity simulations applying a 20% perturbation in anthropogenic emissions in four regions (North America, Europe, South Asia, and East Asia, each defined as a broad rectangle-shaped area) (HTAP, 2010). HTAP showed that the annual average impact of North American emissions on East Asian surface ozone is comparable to the impact of East Asian emissions on North America (0.22 ppb averaged over each rectangular region).

Several studies investigated the influence of transboundary transport on surface ozone over Chinese territory (Wang et al., 2011; Li et al., 2014; Li et al., 2016b; Zhu et al., 2016; Yin et al., 2017). Wang et al. (2011) used tagged ozone simulations with GEOS-Chem to study the global production of surface ozone over China for 2006. They showed that in spring 2006, tropospheric ozone produced over India contributed up to 6 ppb to surface ozone over western China; and that ozone produced over Europe and North America each contributed 2–5 ppb of ozone over northeastern China and North China. Using an emission zero-out method with MOZART simulations (i.e., without versus with emissions), Li et al. (2014) reported that modeled trans-Eurasian ozone

transport enhanced surface ozone over northwestern China by 2–6 ppb in spring 2000.
Using tagged ozone simulations with MOZART, Zhu et al. (2016) revealed significant
springtime ozone transport (~ 6 ppb) from Europe and Africa to Waliguan averaged
from 1997 to 2007 and 3–5 ppb ozone from North and South America together. Using
a tagged ozone method based on the Nested Air Quality Prediction Modeling System
(NAQPMS), Li et al. (2016) found 0.5–3.0 ppb of ozone over northeastern China
produced over the Korean peninsula in 2010. Based on observational and back-
trajectory analyses, Yin et al. (2016) found that ozone at the Nam Co site over Tibet in
spring is greatly affected by anthropogenic contributions from South Asia.
Transboundary ozone due to precursor emissions of a source region can be produced
both within and outside the source region. The two mechanisms contribute roughly
equally for the case of trans-Pacific ozone from East Asia to the western United States
(Zhang et al., 2008; Jiang et al., 2016). And the ozone production along the transport
pathway is largely associated with thermal dissociation of peroxyacetyl nitrate (PAN)
that has been formed in the boundary layer of the NOx emission source region. The
transport of ozone precursors means that ozone produced within a region (from emitted
and transported precursors worldwide) differs from ozone produced from that region's
emissions. This difference affects how ozone over a receptor region is attributed to other
regions (Wang et al., 2011; Li et al., 2014). It is thus important that the contribution of
ozone produced at a "producing region" from emissions of a source region be quantified
explicitly.
Here we simulate the contributions of anthropogenic emissions in individual regions
across the globe to ozone at various heights over China. As typically assumed,
anthropogenic contributions are associated with anthropogenic NOx, carbon monoxide
(CO) and NMVOC emissions, excluding the effect of methane. We use a GEOS-Chem
based two-way coupled modeling system (Yan et al., 2014; 2016) that integrates an
Asian nested model and a global model in a sense of two-way exchange, which better
simulates multi-scale interactions between the nested and global domains. Our study is
focused on spring 2008, in which season a comprehensive set of ground, aircraft and
ozonesonde measurements over China is available for model evaluation. Also,
transboundary transport of ozone is most significant in spring due to active cyclonic
activities and strong westerly winds (Liang et al., 2004; Wang et al., 2011; HATP,
108 2010).

We explicitly identify ozone produced in 10 individual regions of the world from
anthropogenic precursor emissions in each of eight source regions. These 10 producing
regions include the troposphere of the eight emitters, the troposphere of total oceanic
regions, and the stratosphere. For this purpose, we combine the emission zero-out
method and the tagged ozone approach (Wang et al., 1998). The zero-out or similar
emission perturbation methods are widely used to quantify the contribution of
emissions in a source region to a receptor region as a combined result of the two
production-transport mechanisms aforementioned (Lin et al., 2008; HTAP, 2010; Lin
et al 2012a; Li et al., 2014). The tagged ozone approach quantifies the ozone produced
in any designated region with no information about whether the associated precursors
are emitted in that region or are transported from somewhere else (Wang et al., 1998;
Wang et al., 2011; Li et al., 2016b). To account for ozone production nonlinearity, we
use a simple linear weighting method to adjusting simulation results, similar to Li et al.
(2016a).
The rest of our paper is organized as follows. Section 2 presents model simulations,
measurement data, and the ozone source attribution method. Section 3 evaluates the
modeled ozone and CO using ground, aircraft and ozonesonde observations. Section 4
analyzes the modeled contributions to near-surface ozone over China by natural sources
as well as anthropogenic emissions in individual regions. Section 5 shows the ozone
source attribution at different heights of the troposphere. For each emission source
region, it also separates the contribution of ozone produced within that source region
from the contribution produced outside of that source region. Section 6 concludes the
study.
**2. Model simulations, measurements, and source attribution method**
*2.1 Two-way coupled GEOS-Chem modeling system*
The two-way coupled system (Yan et al., 2014; Yan et al., 2016) is built upon version
9-02 of GEOS-Chem (http://wiki.seas.harvard.edu/geos-chem/index.php/Main_Page).
Here we couple the global GEOS-Chem model (at 2.5° long. × 2° lat.) with its
nested model covering Asia (70°E–150°E, 11°S–55°N, at 0.667° long. × 0.5° lat.).
Through the PeKing University CouPLer (PKUCPL) for two-way coupling, for every
three hours the global model provides lateral boundary conditions for the nested
model, while the nested model results replace the global model results within the
nested domain (Yan et al., 2014; 2016). Both models are driven by the GEOS-5
assimilated meteorological fields at respective horizontal resolutions from National
Aeronautics and Space Administration Global Modeling and Assimilation Office.
There are 47 vertical layers for both models, and the lowest 10 layers are about 130 m
thick each.
Both the global and nested GEOS-Chem models include the full gaseous HOx-Ox-
NOx-CO-NMVOC chemistry (Mao et al., 2013) and online aerosol calculations, with
further updates detailed in Lin et al. (2012) and Yan et al. (2016). As aromatics are not
explicitly represented in the model, following Lin et al. (2012), we approximate the
ozone production of aromatics by increasing anthropogenic emissions of propene by a
factor of four, based on their reactivity differences, their similarity in emission spatial
variability, and recently estimated emission amounts of aromatics (Liu et al., 2010). We
use the Linoz scheme for ozone production in the stratosphere (McLinden et al., 2000).
We adjust the stratospheric production rate in the nested model to ensure that the
stratosphere-troposphere exchange (STE) of ozone in the nested model matches the
STE in the global model over the same nested domain (Yan et al., 2016). Vertical

mixing in the planetary boundary layer (PBL) is parameterized by a non-local scheme (Holtslag and Boville, 1993; Lin and McElroy, 2010), and convection in the model employs the relaxed Arakawa-Schubert scheme (Moorthi and Suarez, 1992).

Table 1 lists the emission inventories used here. Global anthropogenic emissions of NOx and CO in 2008 are from the Emission Database for Global Atmospheric Research (EDGAR v4.2). Anthropogenic NMVOC emissions are from the REanalysis of TROpospheric chemical composition (RETRO) inventory for 2000. Anthropogenic emissions over China, the rest of Asia, the United States, Canada, Mexico and Europe are replaced by regional inventories MEIC (for 2008), INTEX-B (for 2006), NEI2005 (for 2005), CAC (for 2008), BRAVO (for 1999) and EMEP (for 2007), respectively. Emissions of CO and NOx are scaled to 2008 in the United States and to 2006 in Mexico. (http://wiki.seas.harvard.edu/geos-chem/index.php/Scale_factors_for_anthropogenic_emissions). We use daily biomass burning emissions from Global Fire Emission Database version 3 (GFED3) (van der Werf et al., 2010). Biogenic emissions of NMVOC are calculated online based on the MEGAN v2.1 scheme (Guenther et al., 2012). For lightning NOx emissions, flash rates are calculated based on the cloud top height and constrained by climatological satellite observations (Murray et al., 2012), and the vertical profile of emitted NOx follows Otto et al. (2010). Online calculation of soil NOx emissions follows Hudman et al. (2012).

*2.2 Zero-out simulations, tagged ozone simulations, and weighted adjustment*

Table 2 presents 10 full-chemistry simulations to quantify Chinese and foreign anthropogenic contributions to springtime ozone over China in 2008. A base simulation (CTL) includes all emissions. The second simulation excludes anthropogenic NOx, CO and NMVOC emissions worldwide to determine the natural ozone (xANTH). Eight additional simulations exclude anthropogenic emissions over China (xCH), Japan and Korea (xJAKO), South-East Asia (xSEA), South Asia (xSA), Rest of Asia (xROA), Europe (xEU), North America (xNA) and Rest of World (xROW), respectively (see regional definitions in Fig. 1). All simulations cover November 2007 through May 2008, with the first four months used for spin-up, except for additional CTL simulations in other years for model evaluation purposes.

Table 2 also shows 10 tagged simulations (denoted as T_CTL, T_xANTH, etc.) with respect to CTL and other eight zero-out sensitivity simulations. Each tagged simulation includes 10 tracers to track ozone produced within the troposphere of eight source regions, produced within the troposphere of the oceanic regions, or transported from the stratosphere. Considering the time for STE of air, all tagged ozone simulations are spun up for 10 years.

Ozone production is nonlinearly dependent on its precursors, adding uncertainties to the source attribution calculated by emission perturbation methods (Wu et al., 2009). To account for this issue, we use a linear weighting method to adjust all ozone attribution results, unless stated otherwise. Below is an example to determine the

contribution from Chinese anthropogenic emissions (here Ci represents the sensitivity
simulation for one of the eight emission source regions). The adjustment is done for
each grid cell over China. Equation 1 calculates the fractional Chinese contribution ($\alpha$)
to the sum of ozone from individual anthropogenic source regions and from natural
sources; the simulations involved are all full-chemistry runs (CTL, xCH, xEU, …,
xANTH). Equation 2 applies the fractional contribution $\alpha$ to the total ozone in CTL to
obtain the final adjusted Chinese contribution.
$$\alpha = \frac{\text{Con(CTL)} - \text{Con(xCH)}}{\sum_{i=1}^{8}[\text{Con(CTL)} - \text{Con(Ci)}] + \text{Con(xANTH)}} \tag{1}$$
$$C_{CH} = \alpha \times \text{Con(CTL)} = \frac{\text{Con(CTL)} - \text{Con(xCH)}}{\sum_{i=1}^{8}[\text{Con(CTL)} - \text{Con(Ci)}] + \text{Con(xANTH)}} \times \text{Con(CTL)} \tag{2}$$
Figure 2a shows the spatial distribution of the ratio of total surface ozone in CTL to
the pre-linear-weighting-adjustment sum of natural ozone, domestic anthropogenic
ozone and foreign anthropogenic ozone. The ratio is close to unity over central and
western China. Over most of the eastern regions, the ratio is between 1.05 and 1.10,
although it can reach 1.30 at a few locations. Figure 2b further compares the vertical
profile of China average total ozone in CTL and the profile of pre-linear-weighting-
adjustment sum of natural ozone, domestic anthropogenic ozone and foreign
anthropogenic ozone. The difference between the two profiles is rather small. These
results suggest relative small effects of chemical nonlinearity. And the linear weighting
adjustment further removes these effects.
A similar approach was used by Li et al. (2016a) to estimate the contribution of China
to global radiative forcing, although in their study 20% (instead of 100%) of emissions
over individual emission source regions are removed in the sensitivity simulations.
*2.3 Measurements*
This study presents model evaluation over China and its neighboring countries in spring.
We also evaluate the simulation of CO, a relatively long-lived transport tracer. Figure
3 shows the suite of ground, aircraft and ozonesonde measurements.
*2.3.1 Surface measurements*
Measurements from a total of 32 ground sites are used here; see Tables 3 and 4 for
geographical information. Routine observations of ozone and CO in China were
scarcely available before 2013. Hourly data are available for this study from five
rural/background sites across China maintained by the Chinese Meteorological
Administration (Xu et al., 2008; Lin et al., 2009; Fang et al., 2014; Ma et al., 2014).
These sites include a rural site (Gucheng over North China Plain), three regional
background sites (Longfengshan over the northeast, Lin'an over the east, and Shangri-
La over the southwest), and a Global Atmosphere Watch (GAW) background site
(Waliguan over the west). Data are available for 2007 at Gucheng and Longfengshan
and for 2008 at other three sites.
We also use hourly ozone and CO measurements in spring 2008 from six GAW
background sites in the vicinity of China from the World Date Center for Greenhouse
Gases (WDCGG, http://ds.data.jma.go.jp/gmd/wdcgg/cgi-bin/wdcgg/catalogue.cgi).
These sites include Issyk-Kul in Kyrgyzstan, Everest-Pyramid in Nepal, Bukit Koto
Tabang in Indonesia, and Yonagunijima, Tsukuba and Ryori in Japan.
To obtain a more comprehensive observation dataset for model evaluation, we further
use monthly mean ozone data in spring 2008 from 15 remote/rural sites from the Acid
Deposition       Monitoring        Network        in       East       Asia       (EANET,
http://www.eanet.asia/product/index.html). We also collect monthly ozone observation
data at six sites over China from the literature, including data at three mountain sites
(Mts. Tai, Hua, and Huang).
*2.3.2 Measurements of vertical profiles*
To evaluate vertical distribution of ozone and CO over China, we use observations from
the Measurements of Ozone and Water Vapor by Airbus In-Service Aircraft (MOZAIC)
program (Marenco et al., 1998). Data during both ascending and descending processes
of the aircrafts are available during spring 2000–2005 at three airports (Beijing,
Shanghai, and Hong Kong). The vertical resolution is 150 m.
We further use the ozonesonde data at six sites in spring 2008 from the World Ozone
and         Ultraviolet         Date         Center         (WOUDC,
http://www.woudc.org/data/explore.php?lang=en) operated by the Meteorological
Service of Canada. The six sites include Hanoi in Vietnam, Hong Kong in China,
Sepang Airport in Malaysia, and Sapporo, NAHA and Tateno in Japan. Ozonesondes
are launched every few days, thus the data are relatively scarce. We also use the GPSO3
ozonesonde data in spring 2008 over Beijing measured by the Institute of Atmospheric
Physics (IAP) of the Chinese Academy of Sciences (Wang et al., 2012). All ozonesonde
measurements were launched at around 14:00 local time.
**3. Model evaluation**
Here we focus on model evaluation over China and its neighboring area in spring.
Global ozone evaluation of the two-way coupled model system is detailed in Yan et al.
(2016) using 1420 ground sites, various aircraft observations and satellite
measurements, although the observations over China are sparse.
*3.1 Surface ozone and CO over China and nearby countries*
Figure 4 compares the springtime time series of modeled (solid red line) and observed
(solid black line) maximum daily average 8-hour (MDA8) ozone concentrations at 10
sites with daily measurements. Model data are sampled at times and locations
coincident with valid observations.
Figure 4a–b evaluates the model results at Gucheng and Longfengshan. To compare to
observations in spring 2007 at these two sites, we conduct an additional full chemistry
simulation for 2007. At these sites, the model captures the observed MDA8 ozone, with
a normalized mean bias (NMB) of 3% at Gucheng and 5% at Longfengshan. The
respective correlation coefficients (R) for day-to-day variability are 0.51 and 0.59; the
modest correlation is primarily because the model does not capture a few short-term
spikes.
At Lin'an (Fig. 4c), the modeled spring average MDA8 ozone matches the observed
value (68.9 ppb versus 65.1 ppb, R = 0.64). The model cannot reproduce the observed
extreme low values on several days. This deficiency is likely due to representative
errors of model meteorology. Located in a hilly area, this site often receives rains and
fogs in spring, which is not captured by the model meteorology at a resolution of 0.667°
long. × 0.5° lat. We find that the extremely low observed ozone values normally occur
on days with high relative humidity (black dashed line, reflecting rainy or foggy days),
when the model underestimates RH (red dashed line) and overestimates ozone.
At Shangri-La, Waliguan and Issyk-Kul (Fig. 4d–f), with high altitudes (1640–3816 m)
and little local anthropogenic sources, the model overestimates the MDA8 ozone by 7–
8 ppb (12–14%). At Everest-Pyramid in Nepal (Fig. 4g, at 5079 m altitude), the
overestimate reaches 13 ppb (19%). These positive biases are due to overestimated
transport from the free troposphere and stratosphere. The model captures the temporal
variability of MDA8 ozone quite well (R = 0.72–0.78) at the three Japanese sites
(Yonagunijima, Tsukuba and Ryori, Fig. 4h–j). Its NMB is within 3% at Yonagunijima
and Ryori. There is an overestimate at Tsukuba (NMB = 19%), mostly reflecting the
large positive biases on a few days.
Table 4 shows model comparisons with monthly mean EANET ozone data. These data
represent daily mean rather than MDA8 values, based on the availability of
observations. At seven sites, the model results exceed the observations with a mean
difference by 7 ppb (16%). At the other eight sites, the model results are smaller than
the observations with a mean difference by 7 ppb (11%). These differences reflect
model biases as well as a sampling bias due to lack of knowledge on which days contain
valid observations.
Table 4 further compares the modeled monthly mean daily mean ozone in spring 2008
to the observations in various years collected from the literature. Again, the comparison
is affected by a sampling bias. Although not our primary focus, this extended
comparison gives a sense of how model ozone is situated in the general ozone pollution
phenomena in China. The model reproduces the average magnitude of ozone at the
three mountainous sites (Mts. Tai, Hua and Huang) with a mean bias below 5 ppb (9%).
The model has a large overestimate by 48% at the Hok Tsui coastal rural site in Hong
Kong (36.0 versus 53.4 ppb), although the times are different (2008 versus 1994–2007).
Wang et al. (2009) shows that the springtime ozone concentration at this site increased
from 1994 to 2007 at a rate of 0.41 ppb/yr, partly explaining this difference. The
remaining difference may reflect that the model resolution is not able to represent the
complex local terrain and land-sea contrast at this site. The model overestimates ozone
at an urban site in Nanjing by 16%, although the observations were made in 2000–2002
when Chinese anthropogenic emissions of NOx were only about half of those in 2008
(Xia et al., 2016).
We also evaluate the modeled daily average CO at six sites within and outside China
with available hourly observations (Fig. 5). Overall, the model captures the day-to-day
variability of daily mean CO fairly well (R = 0.40 at Lin'an, 0.60 at Shangri-La, 0.56
at Ryori, and 0.73–0.82 at other three sites). It has a small mean bias (within 4%) at
Bukit Koto Tabang and Ryori, although with negative biases (by 13–33%) at other four
sites. Such an underestimate is typical in global simulations (Young et al., 2013), and
it may be related to excessive OH (Young et al., 2013; Yan et al., 2014; 2016) and/or
underestimated emissions (Kopacz et al., 2010; Wang et al., 2011). As compared to the
coarse-resolution global model alone, our two-way coupling results in less CO
underestimate (Yan et al., 2014), although it does not eliminate the bias.
*3.2 Vertical profiles of ozone and CO*
Figure 6a–c compares modeled ozone in 2008 to MOZAIC data over 2000–2005 at the
airports of Beijing, Shanghai and Hong Kong. Although model and MOZAIC data are
in different years, to achieve best sampling consistency, we sample the model results at
times of day when the commercial aircrafts take off or land in with available MOZAIC
data. The timing information is shown in Fig. 6. GEOS-Chem reproduces the vertical
gradient of MOZIAC ozone in general. The model underestimates MOZIAC ozone in
the PBL over Beijing Airport mainly due to inconsistent temporal sampling, as further
comparison with GPSO3 ozonesonde data (Bian et al., 2007; Wang et al., 2012), where
model results are sampled at times coincident with the observations, shows little model
bias (within 4%, Fig. 6g). Over Hong Kong, the model captures the weak vertical
gradient between 2 km and 11 km, although it has a positive bias below 2 km due to its
inability to capture the complex terrains and local pollution source characteristics
around the airport. The model overestimates ozone in the middle and upper troposphere
over Shanghai, with larger biases at higher altitudes, likely indicating too strong STE.
Other causes may include differences in meteorology and growth in emissions between
2000–2005 and 2008, as discussed for the surface ozone in Sect. 3.1.
Figure 7 compares the modeled ozone profiles to WOUDC data at six sites. Here model
results are sampled at ozonesonde launch times, and ozonesonde data are regridded to
match the model vertical resolution. Overall, GEOS-Chem captures the vertical
gradient of ozone fairly well. The model reproduces the overall weak vertical gradients
at Hanoi, Hong Kong, Sepang and NAHA. It also reproduces the rapid increases above
8 km at Sapporo and Tateno, although it has positive biases at 10–20 ppb. GEOS-Chem
reproduces the observed middle and upper tropospheric ozone at Hong Kong and
Sepang, although it has an overestimate in the lower troposphere, consistent with the
bias shown in Fig. 6c.
Figure 6d–f also compares the modeled CO with the MOZAIC data. Similar to the
evaluation results for surface CO, GEOS-Chem generally underestimates the MOZAIC
CO at most heights above the three airports, although it captures the vertical shape fairly
well.
*3.3 Summarizing remark on model evaluation*
Our simulation has a small NMB for surface ozone, at about 10% averaged over 10
sites with hourly data (Fig. 4 and Table 3) and about 15% averaged over 21 sites with
monthly data from EANET and the literature (Table 4). The model also captures the
general vertical distribution of ozone at ten places over China and nearby regions, with
a tropospheric mean bias at 12%. These agreements allow using the model for source
attribution studies in the next sections. On the other hand, with a horizontal resolution
of about 50 km over Asia, the model often fails to simulate the complex terrains, local
meteorological conditions, and/or local emission characteristics at several hilly or
airport sites. The model also tends to overestimate the STE influences over Asia.
Addressing these issues warrant future research with improved model resolutions and
STE representation.
GEOS-Chem tends to underestimate CO over Asia (by 20% on average), similar to
many other models (Kopacz et al., 2010; Young et al., 2013). We conduct a sensitivity
simulation by doubling Chinese anthropogenic CO emissions, which result in a slight
increase in surface ozone by 0.1–0.4 ppb and 2–3 ppb over clean and polluted areas of
China, respectively. The low sensitivity of ozone to CO emissions was also found by
Jiang et al. (2015). We thus conclude that our ozone simulations over China are
influenced insignificantly by the underestimate in CO.
**4. Source attribution modeling for surface ozone over China**
*4.1 Total, background and natural ozone*
Figure 8a shows the modeled spatial distribution of near-surface daily mean ozone in
spring 2008 over China from all natural and anthropogenic sources, i.e., the CTL case.
Ozone concentrations reach 75–80 ppb over the southern Tibetan Plateau, and they are
minimum (25–40 ppb) over the North China Plain and many populous cities across
eastern China. Ozone are about 45–60 ppb over the vast southeast, northwest and
northeast.
The simulated natural ozone (i.e., without anthropogenic emissions worldwide, the
xANTH case) shows a strong gradient from the southern Tibetan Plateau (65–75 ppb)
to the northwest (35–40 ppb) and the east (20–35 ppb) (Fig. 8c). Wang et al. (2011)
shows similar gradients of natural ozone in 2006. Natural ozone contributes 80–90% of
total surface ozone over Tibet and the northwest with low local anthropogenic
emissions. The large natural ozone concentrations over Tibet are a result of vertical
transport from the free troposphere and stratosphere due to its high altitudes and hilly
terrains (that are conducive to vertical exchange) (Ding and Wang, 2006;Lin et al.,
2015;Xu et al., 2017). They pose potential threats for public health and ecosystems
there.
The simulated background ozone (i.e., without Chinese anthropogenic emissions, the
xCH case) is shown in Fig. 8b. The background ozone is higher than the natural ozone
by 2–11 ppb over most Chinese regions (Fig. 9b). This indicates large influences of
foreign anthropogenic emissions through atmospheric transport of ozone and its
precursors, as discussed in detail below.
*4.2 Domestic versus foreign anthropogenic contributions to ozone*
Figure 9a shows the spatial distribution of domestic anthropogenic contributions to
daily mean surface ozone over China (difference between the control run and the
sensitivity simulation, CTL − xCH, followed by a linear weighting adjustment). Over
most of the west and northeast, Chinese anthropogenic emissions are relatively low,
and they result in ozone concentrations by 0–4 ppb. In contrast, domestic contributions
reach 16–25 ppb over the south due to more emissions and favorable conditions for
photochemistry. Over the North China Plain and many populous cities, Chinese
anthropogenic emissions lead to reductions (instead of enhancements) of surface ozone.
This is because of a weak ozone production efficiency and a strong titration effect by
excessive domestic NOx emissions. Figure 9d–f shows that when Ox (= $O_3$ + $NO_2$) is
considered, Chinese anthropogenic contributions vary from 2–4 ppb over the west to
6–12 ppb over the North China Plain and to 20–35 ppb over the southeast (Fig. 9d).
Figure 9b shows the simulated contributions to Chinese surface ozone by all foreign
anthropogenic emissions. Foreign contributions reach 7–11 ppb along much of Chinese
borders, and they exceed 6 ppb over the vast northern regions. The foreign contribution
reduces from the border to the inner areas, with a minimum (2–3 ppb) over the Sichuan
Basin where the air is more isolated. In terms of anthropogenic ozone, foreign
contributions account for up to 90% over most of western and northeastern China (Fig.
9c), consistent with the findings by Li et al. (2015) for western China in 2000. Foreign
anthropogenic contributions to Ox over China are similar to their contributions to ozone
(Fig. 9e), except at places with strong Chinese NOx emissions that lead to titration of
ozone.
Figure 10 further shows the contributions to Chinese surface ozone by anthropogenic
emissions in seven individual foreign regions. The pattern of influence differs among
these source regions due to differences in the location of source region, emission
magnitude, pollutant lifetimes and transport pathways. Anthropogenic emissions in
Japan and Korea result in 0.6–2.1 ppb of ozone enhancement along the Chinese coast.
The tagged ozone simulation with NAQPMS by Li et al. (2016) also showed that about
0.5–3.0 ppb of ozone over northeastern China in spring 2010 were produced over Korea
peninsula, although there is a difference between ozone produced over a region and
ozone produced from that region's emissions. Emissions from South-East Asia
contribute 1–5 ppb over much of the southern provinces. Emissions from South Asia
mostly affect southwestern China and Tibet (by up to 5–10 ppb over the border), due to
effective transport by strong southwesterly associated with the Indian Monsoon. The
"Rest of Asia" consists of many countries to the west of China, whose total
contributions are about 2–5 ppb over much of northwestern China.
European anthropogenic emissions contribute 2.1–3.0 ppb of ozone along the northern
border of China. The contributions decrease southward, and are above 1 ppb over half
of Chinese land areas. The Model for Ozone and Related chemical Tracers (MOZART)
simulations by Li et al. (2015) also showed a European contribution by 2 ppb to surface
ozone over North China in 2000. North American anthropogenic emissions increase
ozone by 1.8–2.7 ppb over much of western China, by 1.5–2.1 ppb over the populous
North China Plain, and by less than 0.9 ppb over the south. The contributions are
smaller than springtime Asian anthropogenic influences on western North America
(e.g., 1–5 ppb averaged over 2001–2005 (Brown-Steiner and Hess, 2011b)), although
the affected population is larger by roughly an order of magnitude.
Influences from "Rest of World" are about 0.6–1.2 ppb over Tibet and smaller over
other Chinese land territory. The larger values over Tibet reflect its higher altitude and
greater sensitivity to long-range transport via the free troposphere.
Figure 11a shows whether domestic or foreign anthropogenic contributions are higher
at individual locations. Domestic anthropogenic contributions are higher than foreign
contributions over southern China and parts of northern China. However, foreign
anthropogenic contributions exceed domestic contributions over western China and
most of the north, including the populated North China Plain. Over western China,
foreign emissions contribute 70–90% of the total anthropogenic ozone (Fig. 9c).
Figure 11b further highlights the largest foreign contributor to surface anthropogenic
ozone at each location of China. North America is the largest foreign contributor over
about half of Chinese land territory, including the populated North China Plain. Europe
is the largest foreign contributor for the vast northeastern region, Rest of Asia for the
western border region, South Asia for southwestern China, South-East Asia for
southern China, and Japan and Korea for the eastern coast of China.
*4.3 Discussion of source attribution with an alternative 20% perturbation method, on*
*extreme ozone, and on other years*
The HTAP and several other studies have used 20% perturbation simulations (i.e.,
reducing anthropogenic emissions in each source region by 20%) to study the
transboundary ozone problem. Such studies are source-receptor analyses that are more
relevant to the question of how much a modest cut in foreign emissions would reduce
ozone pollution over a targeted receptor region. To compare with such a method, here
we ran one more set of full chemistry simulations by decreasing 20% anthropogenic
emissions over each of the eight emission source regions (see the detailed information
in Table A2). We also applied the linear weighting method to account for the non-
linearity of ozone chemistry. Figures 9a and 12a compare the Chinese anthropogenic
contributed ozone calculated from zero-out and from 20%-perturbation simulations.
Compared to the zero-out method, the 20% perturbation method leads to less Chinese
contributed ozone, with negative values over more regions and smaller positive values
over southern China. This result confirms our general finding that in spring 2008, the
excessive domestic NOx emissions lead to relatively weak ozone production and/or
strong ozone titration. Comparing with the zero-out method, the absolute foreign
anthropogenic ozone obtained from 20%-perturbation simulations are smaller by 2–3
ppb over the northern border of China (comparing Figs. 9b and 12b), whereas the
percentage foreign contributions increase from 10–20% to 20–40% over southeastern
China (comparing Fig 9c and 12c). Nonetheless, the spatial patterns are similar
between the two methods for both the absolute and the relative foreign contributions.
As peak ozone is a critical problem for human health, here we show the domestic versus
foreign contributions to modeled extreme ozone values in spring 2008 (defined as the
average of the top 5% hourly ozone concentrations) (Fig. 12d–f). As expected, Chinese
domestic contribution is larger for extreme ozone than for mean ozone; the negative
values also disappear over North China Plain and Northeast China (comparing Fig. 9a
and 12d). The absolute foreign contribution (in ppb) is also enhanced across China
(comparing Fig. 9b and 12e). The percentage foreign contribution is within 10% over
southern China, about 10–50% over the north, and above 70% over the west.
Nevertheless, these results for extreme ozone should be interpreted with more caution,
as the model cannot simulate the dates of extreme ozone very well (Fig. 4).
Previous studies have shown notable interannual variability in surface ozone over
China driven by changes in precursor emissions and meteorology (Xu et al., 2008; Jin
et al., 2015; Wang et al., 2017). To test how the interannual variability of meteorology
and emissions would affect our source attribution findings, we have repeated all zero-
out runs for spring 2012, the latest year when the GEOS-5 meteorological fields are
available. Emissions for 2012 were adopted from the Community Emissions Data
System (CEDS) inventory (Hoesly et al., 2018); 2012 is also the latest year the CEDS
emissions for China are adjusted by the MEIC inventory. Table 5 shows the
anthropogenic emissions in the two years. All zero-out simulation results in 2012
underwent the same linear weighting adjustment as for those in 2008. Figure 12g–i
show the results for domestic versus foreign contributed ozone in spring 2012, as
compared to the results for spring 2008 (Fig. 9a–c). In absolute terms, Chinese
contributed ozone are similar between 2008 and 2012 (comparing Fig. 12g and Fig.
9a), reflecting the slight changes in domestic precursor emissions (Table 5). From 2008
to 2012, the absolute foreign contributed ozone increase along the southern boarder
due to much enhanced emissions in South-East Asia and South Asia. The absolute
foreign contributions decrease over the north and south, reflecting the net effect of
changes in European and North American emissions (within 20% for both NOx and
NMVOC), increased emissions in Rest of Asia, and changes in meteorology. In relative
terms (Figs. 9c and 12i), the percentage foreign anthropogenic contributions to total
anthropogenic ozone decrease from 2008 to 2012 over southern China. Nonetheless,
in both years the percentage foreign contributions exceed 50% over western China and
are 5–40% over southern China. Therefore our general finding that both foreign and
domestic contributions to Chinese anthropogenic ozone are important holds true for
these two years.
**5. Vertical distributions of domestic and foreign anthropogenic contributions**
Figure 13a shows the domestic and foreign anthropogenic contributions to daily mean
ozone at different heights above the ground averaged over China. The black line shows
that Chinese emissions contribute 6.0–10.5 ppb of ozone below 2 km over China, with
a maximum value at 0.7 km. This average amount of contribution reflects compensation
between positive values over most regions and negative values over the North China
Plain and many populous cities (see Sect. 4.2). Above 0.7 km, Chinese contribution
decreases rapidly until 3 ppb at 5 km, above which height the contribution declines
slowly until a value at 1 ppb at 12 km. By comparison, Chinese contribution to Ox is
about 7–11 ppb below 2 km, and at higher altitudes the contribution is almost identical
to that for ozone (not shown). The small contributions above 2 km for both ozone and
Ox are because as ozone and precursors associated with Chinese emissions are lifted to
higher altitudes, they are transported out of Chinese territory and destroyed gradually.
The grey line in Fig. 13a shows that the total foreign contribution is about 5.2–7.8 ppb
at different heights with a reverse "C" shape, i.e., higher values at 3–9 km and lower
values above or below that layer. The foreign contribution exceeds Chinese
contribution at all heights above 2 km. Nonetheless, the total (Chinese + foreign)
anthropogenic ozone is less than one third of natural ozone throughout the troposphere.
Figure 11c shows that of ozone over China produced from all anthropogenic emissions,
foreign emissions together contribute 50% at the surface, 40% at 0.7 km as a minimum,
and 85% in the upper troposphere.
Figure 13b specifies the contribution of each foreign emission source region. Figure
13c further separates the portion of ozone produced within each source region's
territory from the portion produced outside of that source region; results here were
derived from a combination of zero-out simulations (e.g., CTL and xEU) and tagged
simulations (e.g., T_CTL and T_xEU). South-East Asian contribution is about 0.5–2.5
ppb averaged over China, and it increases with height due to strong upwelling that lifts
pollutants to the middle and upper troposphere. The contribution from Japan and Korea
is below 0.5 ppb throughout the troposphere averaged over China (Fig. 13b). The share
of transboundary ozone produced within South-East Asian territory and transported to
China is about 10–45% (mostly below 30%), and the share for ozone produced within
Japan and Korea is even smaller (5–25%) (Fig. 13c), highlighting the importance of
ozone produced by precursors transported out of these two emission source regions.

South Asian contribution is only about 0.5–1.2 ppb throughout the troposphere (Fig. 13b). Although South Asia has more anthropogenic emissions than South-East Asia (Table 2), its contribution to ozone over China is smaller due to blocking of transport by the Himalayas with high elevation (Fig. 3). In addition, the share of transboundary ozone produced within South Asian territory reaches 70–90% below 6 km but declines rapidly to 28% at 12 km (Fig. 13c), a characteristic drastically different from the share for South-East Asia.

The contribution from Rest of Asia is below 1.8 ppb at all heights with a negative vertical gradient (Fig. 13b). Above 3 km, the portion of transboundary ozone produced within the territory of Rest of Asia is similar to that for South Asia (Fig. 13c). However, the portion exhibits a strong vertical gradient below 3 km, with a minimum value at 45% near the ground.

European contribution declines from 1.5 ppb in the lower troposphere to 0.2 ppb at 12 km, similar to that for Rest of Asia (Fig. 13b). In spring, Eurasian frontal activities transport and gradually lift European pollutants to downwind areas. The portion of transboundary ozone produced within European territory is about 55–65% at 3–10 km but is as low as 20% below 1 km (Fig. 13c), suggesting that most Europe-contributed near-surface ozone over China are produced from precursors transported out of Europe.

Figure 13b shows that North American anthropogenic emissions contribute about 1.5–2.5 ppb of ozone below 8 km, although the contribution declines rapidly to 0.2 ppb at 12 km. Compared to Europe, North America is further away from China, but its pollutants can be transported via the strong mid-latitude westerly. Averaged over China, North American contribution is larger than European contribution at all heights, e.g., by a factor of two in the middle and upper troposphere. The higher contribution is due to much more anthropogenic emissions in North America than in Europe. Table 3 shows that North America emits NMVOC nearly twice as much as Europe does; and Wu et al. (2009) showed that the amount of transboundary ozone is nearly proportional to NMVOC emissions of the source region. In addition, Fig. 13c shows that the portion of transboundary ozone produced within North American territory is only about 5–20% below 8 km, reflecting the dominant contribution by ozone produced from transported precursors. The low share of ozone produced within North America is primarily because most of such ozone is destroyed during the transport from North America to China (for about two weeks), given the tropospheric lifetime of ozone at about three weeks (Yan et al., 2016).

The grey line in Fig. 13c shows the average portion of transboundary ozone from all foreign source regions that is produced within the territories of respective foreign regions. The average portion is less than 50% throughout the troposphere, is about 40% at 2 km, and is as low as 25% near the surface. This again highlights the dominant importance of ozone production along with the transport of precursors.

Figure 14 further shows the vertical profiles of ozone from different sources averaged
over regions where Chinese anthropogenic emissions contribute more surface ozone
than total foreign anthropogenic emissions (i.e., southern China, Fig. 14a, b), as well
as averaged over regions where foreign anthropogenic emissions dominate (Fig. 14c,
d). Even over areas where domestic contributions to near-surface ozone exceed total
foreign contributions, the regional average ozone contributed by foreign emissions
exceeds those contributed by domestic emissions above 3.5 km (Fig. 14a). Figure 14b
and d further shows that the (relative) vertical shape of regional average ozone
contributed by each foreign source region is similar to the shape of China averaged
results in Fig. 13b, although the absolute values (in ppb) are different.

## 599 6. Conclusions

This study uses a GEOS-Chem based two-way coupled modeling system to simulate
Chinese and foreign anthropogenic contributions to springtime ozone at different
heights over China. Anthropogenic contributions are associated with anthropogenic
NOx, CO and NMVOC emissions, excluding the effect of methane. We combine the
zero-out simulations and tagged ozone simulations to separate the transboundary ozone
produced within the territory of each emission source region from the ozone produced
by anthropogenic precursors transported out of that source region. We use a weighting
approach to accounting for the effect of nonlinear ozone chemistry on source attribution
estimates. Model evaluation using a suite of ground, aircraft and ozonesonde
measurements show an overall small bias for ozone near the surface and in the
troposphere (10% at 10 surface sites with hourly measurements, 15% at 21 surface sites
with monthly observations, and 12% for vertical profiles). The model underestimates
CO by 20% on average over China and nearby areas, which however does not affect
the simulated ozone significantly.
Model simulations reveal that both total and natural ozone near the surface over China
show a decreasing gradient from the southern Tibetan Plateau to the northwest and the
east. Natural ozone contributes 80–90% of total surface ozone over Tibet and the
northwest with low local anthropogenic emissions. Chinese anthropogenic emissions
enhance surface ozone concentrations by 0–4 ppb over most of the west and northeast
due to low emissions and by 16–25 ppb over the south due to more emissions and
chemically conducive conditions. Chinese anthropogenic emissions result in reduced
ozone, albeit with enhanced Ox, over the North China Plain and many populous cities,
as a result of weak ozone production efficiency and strong titration by excessive
Chinese NOx emissions.
Near the surface, foreign anthropogenic emissions contribute 2–11 ppb of Chinese
ozone, with peak contributions at 7–11 ppb over the border and coastal regions of China.
Over western and northeastern China, foreign emissions account for up to 90% of ozone
of anthropogenic origin. Anthropogenic emissions in Japan and Korea result in 0.6–2.1
ppb of ozone along the Chinese coast. Emissions in South-East Asia contribute 1–5 ppb
over much of southeastern China. South Asian emissions mostly affect southwestern

China and Tibet (by up to 5 ppb), due to effective transport by strong southwesterly associated with the Indian Monsoon. European anthropogenic emissions contribute 2.1–3 ppb along the northern border of China and the contribution decreases southwards. North American anthropogenic emissions increase ozone by 1.8–2.7 ppb over much of the west, by 1.5–2.1 ppb over the populous North China Plain, and by less than 0.9 ppb over the south.

Vertically, for ozone of anthropogenic origin averaged over China, Chinese emissions contribute ~ 6 ppb (50%) of ozone at the surface, 6.0–10.5 ppb below 2 km, decreasing to 3 ppb at 5 km and 1 ppb at 12 km. The total foreign contribution increases from 40–50% below 2 km to 50–85% above that height. The contribution from Japan and Korea is below 0.5 ppb throughout the troposphere averaged over China. Despite its large emissions, South Asia contributes only about 0.5–1.2 ppb throughout the troposphere due to blocking of transport by the Himalayas. South-East Asian contribution increases with height due to strong upwelling that lifts pollutants to the upper troposphere. On the contrary, European contributions decreases from 1.5 ppb in the lower troposphere to 0.2 ppb at 12 km. Despite the long transport distance, North American contribution reaches as much as 1.5–2.5 ppb below 8 km due to its large anthropogenic emissions and the strong mid-latitude westerly favorable for transboundary transport.

For ozone of foreign anthropogenic origin averaged over China, the portion of transboundary ozone produced within foreign source regions is less than 50% throughout the troposphere, albeit with a strong vertical variability, indicating the importance of ozone produced by precursors transported out of those source regions. The portion also differs among each foreign source region of South-East Asia (10–45%) and Japan and Korea (5–25%), South Asia (from 70–90% below 6 km to 28% at 12 km), Europe (from 20% below 1 km to 55–65% at 3–10 km), and North America (5–20% below 8 km). Thus, tracing ozone produced within the territory of a particular region is drastically different from tracing ozone associated with emissions in that region.

In summary, although China is a major pollutant emitter, the ozone above its territory consists primarily of natural sources, especially over western China with low local anthropogenic emissions. Moreover, for ozone of anthropogenic origin, a large portion results from foreign emissions, as analyzed here for spring 2008. In more recent years, Chinese anthropogenic NOx emissions have undergone a rapid decline as a result of domestic emission control (Xia et al., 2016), along with continuous reductions in North America and Western Europe (Yan et al., 2018a; 2018b) and changes in other regions. Future research is needed to quantify the resulting changes in ozone and its geographical origin. In addition, this study does not account for that a substantial portion of anthropogenic emissions in any region are associated with economic production for foreign consumption (Lin et al., 2014; Jiang et al., 2015a), which would affect how pollution is attributed to individual producing or consuming regions (Guan et al., 2014; Lin et al., 2016; Zhang et al., 2017). Nevertheless, our study suggests the great importance of global collaboration on emission reduction to mitigate ozone

pollution in addition to domestic emission control efforts.

## Acknowledgments

This research is supported by the National Natural Science Foundation of China
(41775115) and the 973 program (2014CB441303). We acknowledge Dr. Chen
Hongbin's team at IAP LAGEO for providing the GPSO3 ozonesonde data. We
acknowledge    the    free    use    of    ozone    data    from    WDCGG
(http://ds.data.jma.go.jp/gmd/wdcgg/),                             EANET
(http://www.eanet.asia/product/index.html),                         WOUDC
(http://www.woudc.org/data/explore.php?lang=en),    and    MOZAIC-IAGOS
(http://www.iagos.fr/web/). We thank the European Commission for the support to the
MOZAIC project (1994–2003) and the preparatory phase of IAGOS (2005–2012)
partner institutions of the IAGOS Research Infrastructure (FZJ, DLR, MPI, KIT in
Germany, CNRS, CNES, Météo-France in France and University of Manchester in
United Kingdom), ETHER (CNES-CNRS/INSU) for hosting the database, the
participating airlines (Lufthansa, Air France, Austrian, China Airlines, Iberia, Cathay
Pacific) for the transport free of charge of the instrumentation.

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

Table 1. Emissions used in the model.

| Region | Inventory | Resolution [a] | Year | Species [b] | References & Notes |
|---|---|---|---|---|---|
| Anthropogenic emissions | | | | | |
| Global | EDGAR v4.2 | 0.1° x 0.1°, monthly | 2008 | NOx, SO2, CO, NH3 | http://edgar.jrc.ec.europa.eu/overview.php?v=42 |
| Global | BOND | 1° x 1°, monthly | 2000 | BC and OC | Bond et al. (2007) |
| Global | RETRO | 0.5° x 0.5°, monthly | 2000 | NMVOC | ftp://ftp.retro.enes.org/pub/emissions/aggregated/anthro/0.5x0.5/2000/ |
| Global | ICOADS, shipping | 1° x 1°, monthly | 2002 | NOx, SO2, CO | Wang et al. (2008); http://coast.cms.udel.edu/GlobalShipEmissions/ |
| Global | AEIC, aircraft | 1° x 1°, annual | 2005 | NOx, SO2, CO, NMVOC, BC, OC | Simone et al. (2013) |
| Asia | INTEX-B | 1° x 1°, monthly | 2006 | NOx, SO2, CO, NMVOC, BC, OC, NH3 | Zhang et al. (2009). NH3 only available for 2000. |
| China | MEIC | 0.25° x 0.25°, monthly | 2008 | NOx, SO2, CO, NMVOC, NH3 | Li et al. (2017); Geng et al. (2017); http://www.meicmodel.org/. |
| United States | NEI2005 | 4km x 4km, monthly & weekend/weekday | 2005 [c] | NOx, SO2, CO, NMVOC, NH3, BC, OC | ftp://aftp.fsl.noaa.gov/divisions/taq/emissions_data_2005 |
| Canada | CAC | 1° x 1°, annual | 2008 | NOx, SO2, CO, NH3 | http://www.ec.gc.ca/pdb/cac/cac_home_e.cfm |
| Mexico | BRAVO | 1° x 1°, annual | 1999 [c] | NOx, SO2, CO | Kuhns et al. (2005) |
| Europe | EMEP | 1° x 1°, monthly | 2007 | NOx, SO2, CO | Auvray and Bey (2005); http://www.emep.int/index.html |
| Biomass burning emissions | | | | | |
| Global | GFED3 | 0.5° x 0.5°, daily | 2008 | NOx, SO2, CO, NMVOC, NH3, BC, OC | van der Werf et al., 2010; http://www.globalfiredata.org |
| Natural/Semi-natural emissions (online calculation) | | | | | |
| Global | MEGAN v2.1 | Model resolution | 2008 | ISOP, monoterpenes, sesquiterpenes, MOH, ACET, ETOH, CH2O, ALD2, HCOOH, C2H4, TOLU, PRPE | Guenther et al. (2012) |
| Global | Soil NOx | Model resolution | 2008 | NO | Hudman et al. (2012) |
| Global | Lightning NOx | Model resolution | 2008 | NO | Murray et al. (2012) |

a. Before re-gridded to model horizontal resolutions. For more information, see
http://wiki.seas.harvard.edu/geos-chem/index.php/Anthropogenic_emissions.
b. Notes for NMVOC: RETRO includes PRPE, C3H8, ALK4, ALD2, CH2O and
MEK; in the CTM, MEK emissions are further allocated to MEK (25%) and ACET
(75%). AEIC, INTEX-B and MEIC include PRPE, C2H6, C3H8, ALK4, ALD2,
CH2O, MEK and ACET. NEI05 includes PRPE, C3H8, ALK4, CH2O, MEK and
ACET. EMEP includes PRPE, ALK4, ALD2 and MEK. Emissions of C2H6 outside
Asia are from Xiao et al. (2008).
c. Over the United States and Mexico, emissions of CO, NOx are scaled to 2008 and
2006 respectively. (http://wiki.seas.harvard.edu/geos-
chem/index.php/Scale_factors_for_anthropogenic_emissions).
Table 2. Model simulations.

| Full chemistry simulation | Description | Tagged ozone simulation | Description |
|---|---|---|---|
| CTL | Full-chemistry simulation with all emissions | T_CTL | Driven by daily ozone production and loss rate archived from CTL |
| xANTH | Without global anthropogenic emissions | T_xANTH | With respect to xANTH |
| xCH | Without anthropogenic emissions of China | T_xCH | With respect to xCH |
| xJAKO | Without anthropogenic emissions of Japan and Korea | T_xJAKO | With respect to xJAKO |
| xSEA | Without anthropogenic emissions of South-East Asia | T_xSEA | With respect to xSEA |
| xSA | Without anthropogenic emissions of South Asia | T_xSA | With respect to xSA |
| xROA | Without anthropogenic emissions of Rest of Asia | T_xROA | With respect to xROA |
| xEU | Without anthropogenic emissions of Europe | T_xEU | With respect to xEU |
| xNA | Without anthropogenic emissions of North America | T_xNA | With respect to xNA |
| xROW | Without anthropogenic emissions of Rest of World | T_xROW | With respect to xROW |


Table 3. Comparison of simulated and observed springtime MDA8 ozone and CO at
five regional background sites in China and six global background stations nearby
China with hourly measurements.

| Country | Site | Location | Year | MDA8 ozone | | | CO | | | References & Notes |
|---|---|---|---|---|---|---|---|---|---|---|
| | | | | Obs | Model | NMB | Obs | Model | NMB | |
| | | | | (ppb) | (ppb) | (%) | (ppb) | (ppb) | (%) | |
| | Gucheng | 39.1°N, 115.7°E, 15m | 2007 | 48.8 | 50.2 | 2.9 | | | | Lin et al., 2009 |
| | Longfengshan | 44.7°N, 127.6°E, 331m | 2007 | 50.6 | 52.9 | 4.5 | 290 | 251 | -13.4 | |
| China | Lin'an | 30.2°N, 119.7°E, 132m | 2008 | 65.1 | 68.9 | 5.8 | 628 | 418 | -33.4 | Xu et al., 2008 |
| | Shangri-La | 28.0°N, 99.4°E, 3580m | 2008 | 61.4 | 68.7 | 11.9 | 181 | 139 | -23.2 | Ma et al., 2014 |
| | Waliguan | 36.3°N, 100.9°E, 3816m | 2008 | 56.5 | 64.4 | 14.0 | | | | Xu et al., 2016 |
| Kyrgyzstan | Issyk-Kul | 42.6°N, 77.0°E, 1640m | 2008 | 52.8 | 59.0 | 11.7 | | | | |
| Nepal | Everest-Pyramid | 28.0°N, 86.8°E, 5079m | 2008 | 66.3 | 79.1 | 19.3 | | | | |
| Indonesia | Bukit Koto Tabang | 0.2°S, 100.3°E, 865m | 2008 | | | | 141 | 146 | 3.5 | http://ds.data.jma.go.jp/gmd/wdcgg/cgi-bin/wdcgg/catalogue.cgi |
| | Yonagunijima | 24.5°N, 123.0°E, 30m | 2008 | 54.8 | 56.4 | 2.9 | 208 | 157 | -24.5 | |
| Japan | Tsukuba | 36.1°N, 140.1°E, 25m | 2008 | 47.2 | 56.0 | 18.6 | | | | |
| | Ryori | 39.0°N, 141.8°E, 260m | 2008 | 54.6 | 54.7 | 0.2 | 211 | 203 | -3.8 | |


Table 4. Comparison of simulated springtime monthly mean ozone with observations

from EANET and literature.

| Country | Site | Year | Location | Characteristics | Obs (ppb) | Model (ppb) | NMB (%) | References & Notes |
|---|---|---|---|---|---|---|---|---|
| Japan (EANET) | Rishiri | 2008 | 45.5°N, 141.2°E, 40m | Remote | 55.0 | 46.0 | -16.5 | |
| | Ochiishi | 2008 | 43.1°N, 145.5°E, 49m | Remote | 48.4 | 46.7 | -3.6 | |
| | Tappi | 2008 | 41.3°N, 140.4°E, 105m | Remote | 66.2 | 48.8 | -26.2 | |
| | Sado-seki | 2008 | 38.2°N, 138.4°E, 136m | Remote | 61.3 | 53.3 | -13.0 | |
| | Happo | 2008 | 36.7°N, 137.8°E, 1850m | Remote | 62.0 | 53.8 | -13.2 | |
| | Ijira | 2008 | 35.6°N, 136.7°E, 140m | Rural | 30.7 | 47.8 | 55.7 | |
| | Oki | 2008 | 36.3°N, 133.2°E, 90m | Remote | 58.8 | 55.7 | -5.3 | http://www.eanet.asia/product/index.html |
| | Banryu | 2008 | 34.7°N, 131.8°E, 53m | Urban | 48.5 | 52.1 | 7.5 | |
| | Yusuhara | 2008 | 33.4°N, 132.9°E, 790m | Remote | 53.7 | 53.1 | -1.1 | |
| | Hedo | 2008 | 26.9°N, 128.3°E, 60m | Remote | 53.6 | 54.2 | 1.1 | |
| | Ogasawara | 2008 | 27.1°N, 142.2°E, 230m | Remote | 37.9 | 41.1 | 8.3 | |
| Republic of Korea (EANET) | Kanghwa | 2008 | 37.7°N, 126.3°E, 150m | Rural | 52.3 | 47.4 | -9.4 | |
| | Cheju | 2008 | 33.3°N, 126.2°E, 72m | Remote | 56.3 | 57.7 | 2.5 | |
| | Imsil | 2008 | 35.6°N, 127.2°E | Rural | 30.3 | 48.2 | 58.8 | |
| Russia (EANET) | Mondy | 2008 | 51.7°N, 101.0°E, 2000m | Remote | 43.0 | 49.2 | 14.4 | |
| China (literature) | Miyun | 2006 | 40.5°N, 116.8°E, 152m | Rural | 48.7 | 35.3 | -27.4 | Wang et al. (2011) |
| | Mt. Tai | 2004 | 24.25°N, 117.10°E, 1533m | Rural | 57.0 | 54.8 | -3.9 | |
| | Mt. Hua | 2004 | 34.49°N, 110.09°E, 2064m | Rural | 50.0 | 51.8 | 3.5 | Li et al. (2007) |
| | Mt. Huang | 2004 | 30.13°N, 118.15°E, 1836m | Rural | 59.3 | 54.0 | -9.0 | |
| | Hok Tsui, HongKong | 1994-2007 | 22.2°N, 114.2°E, 60m | Rural | 36.0 | 53.4 | 48.2 | Wang et al. (2009) |
| | Nanjing | 2000-2002 | 32.1°N, 118.7°E | Urban | 27.0 | 31.3 | 16.0 | Tu et al. (2007) |

Table 5. Springtime anthropogenic emissions of NOx, CO and NMVOC in 2008 and
2012 in each source region defined in Fig. 1.

| 2008 | China | Japan and Korea | South-East Asia | South Asia | Rest of Asia | Europe | North America | Rest of world |
|---|---|---|---|---|---|---|---|---|
| NOx (TgN) | 2.0 | 0.3 | 0.4 | 0.4 | 0.7 | 1.2 | 1.3 | 1.0 |
| CO (Tg) | 42.3 | 1.7 | 10.9 | 16.7 | 10.0 | 12.5 | 17.7 | 25.5 |
| NMVOC (TgC) | 2.9 | 0.2 | 1.3 | 1.3 | 1.1 | 1.1 | 2.1 | 1.9 |
| 2012 | | | | | | | | |
| NOx (TgN) | 2.2 | 0.3 | 0.6 | 1.3 | 1.0 | 1.0 | 1.1 | 1.5 |
| CO (Tg) | 39.2 | 2.4 | 15.4 | 21.3 | 8.9 | 7.9 | 13.1 | 38.0 |
| NMVOC (TgC) | 3.0 | 0.2 | 3.0 | 2.4 | 2.3 | 1.2 | 1.8 | 6.8 |

















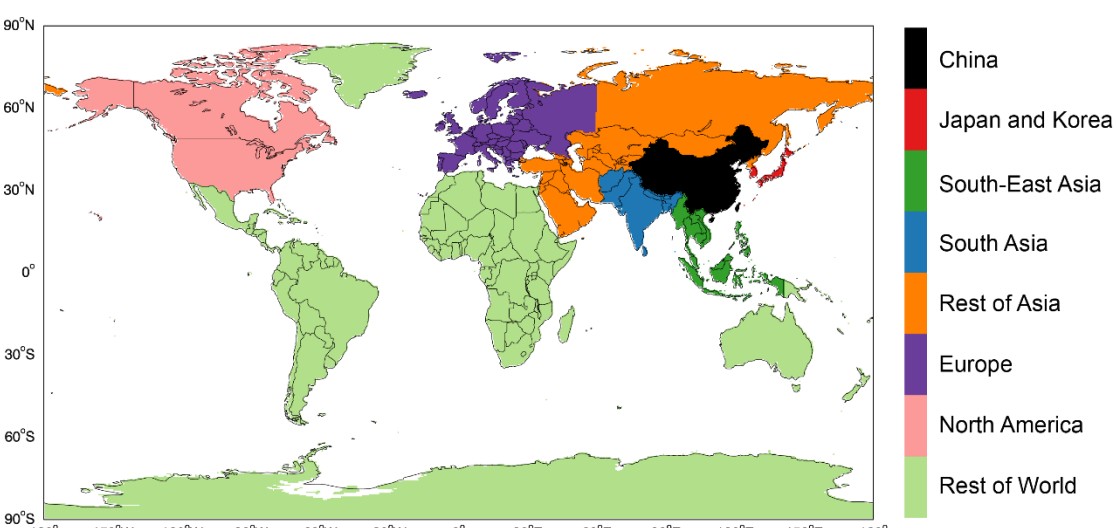
Figure 1. Eight emission source regions.
















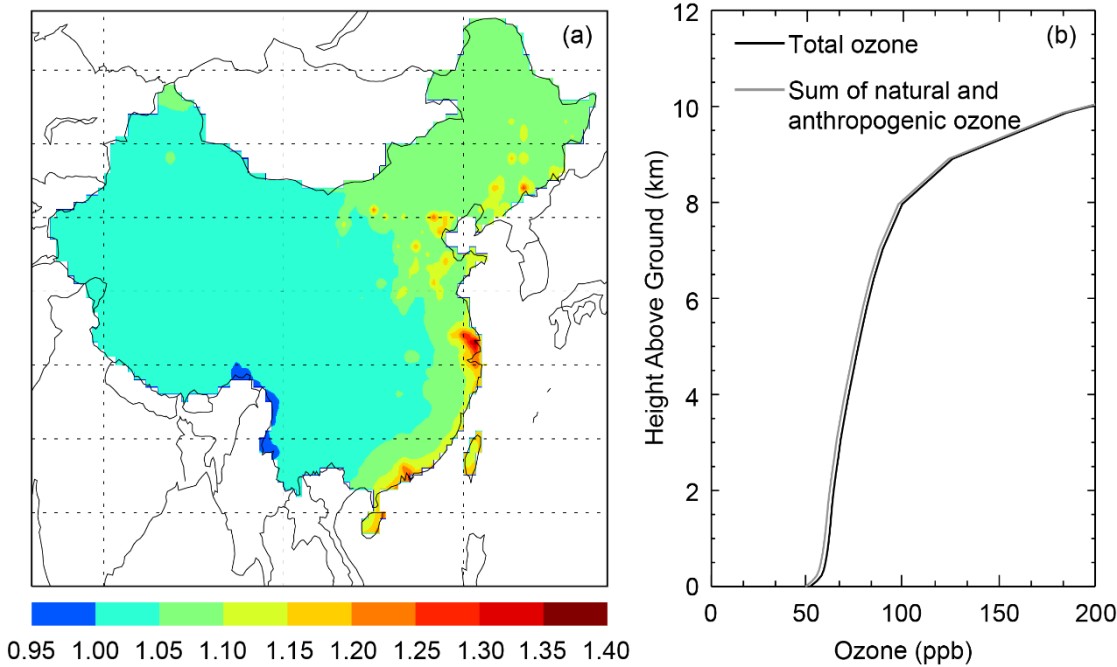


Figure 2. (a) Spatial distribution of the ratio of total surface ozone in CTL to the pre-
linear-weighting-adjustment sum of natural ozone, domestic anthropogenic ozone and
foreign anthropogenic ozone; (b) Vertical profile of China average total ozone from
CTL and the profile of pre-linear-weighting-adjustment sum of natural ozone,
domestic anthropogenic ozone and foreign anthropogenic ozone.

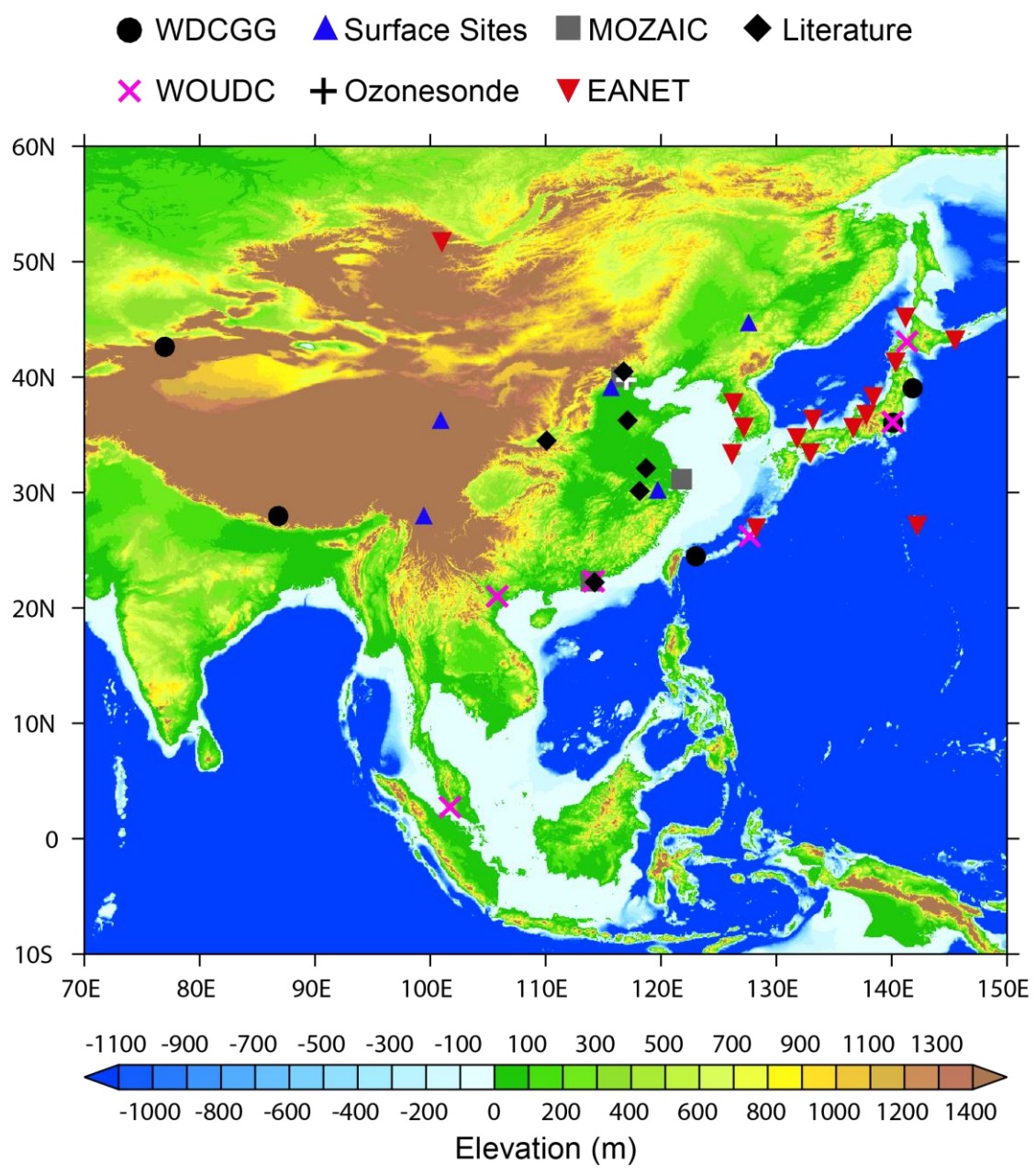


Figure 3. Observation sites overlaying upon the surface elevation map from the 2 min
Gridded Global Relief Data (ETOPO2v2) available at NGDC Marine Trackline
Geophysical database (http://www.ngdc.noaa.gov/mgg/global/etopo2.html).




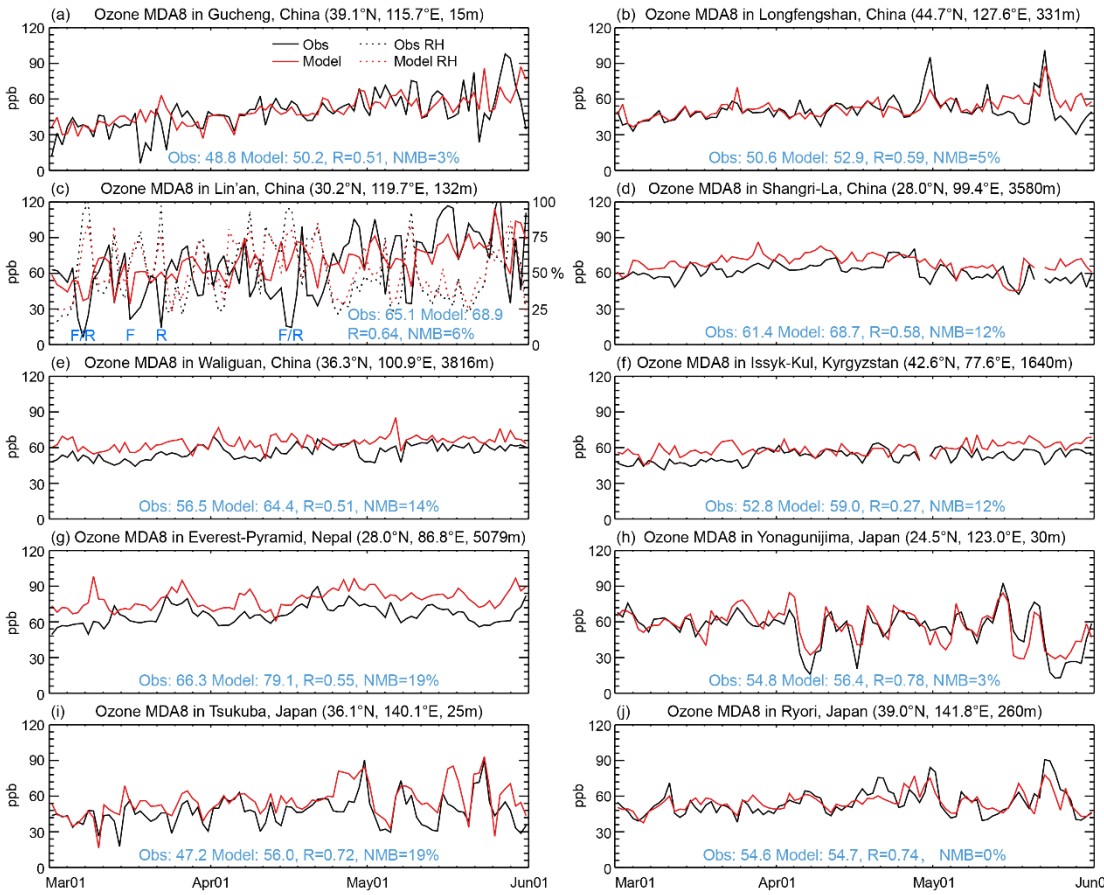


Figure 4. Time series of springtime MDA8 ozone at surface sites over (a–e) China and (f–j) nearby countries. Due to lack of measurement data in 2008, comparisons at Gucheng and Longfengshan are based in 2007. In (c), observed and modeled RH are also compared; and the "F" and "R" symbols denote observed frog or rain, respectively.









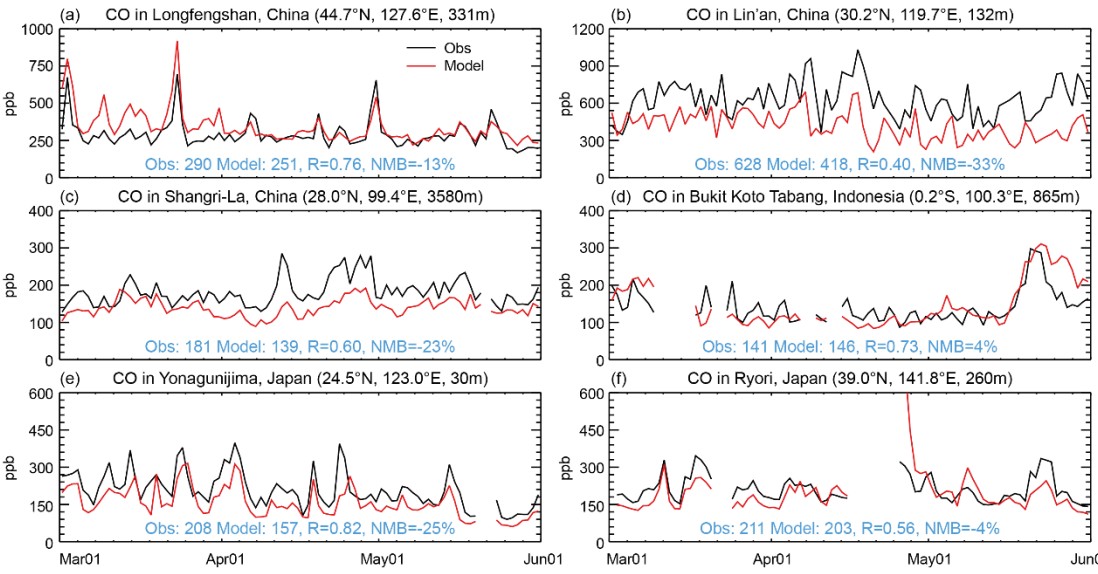


Figure 5. Time series of daily mean CO at six surface sites over (a–c) China and (d–f) nearby countries.













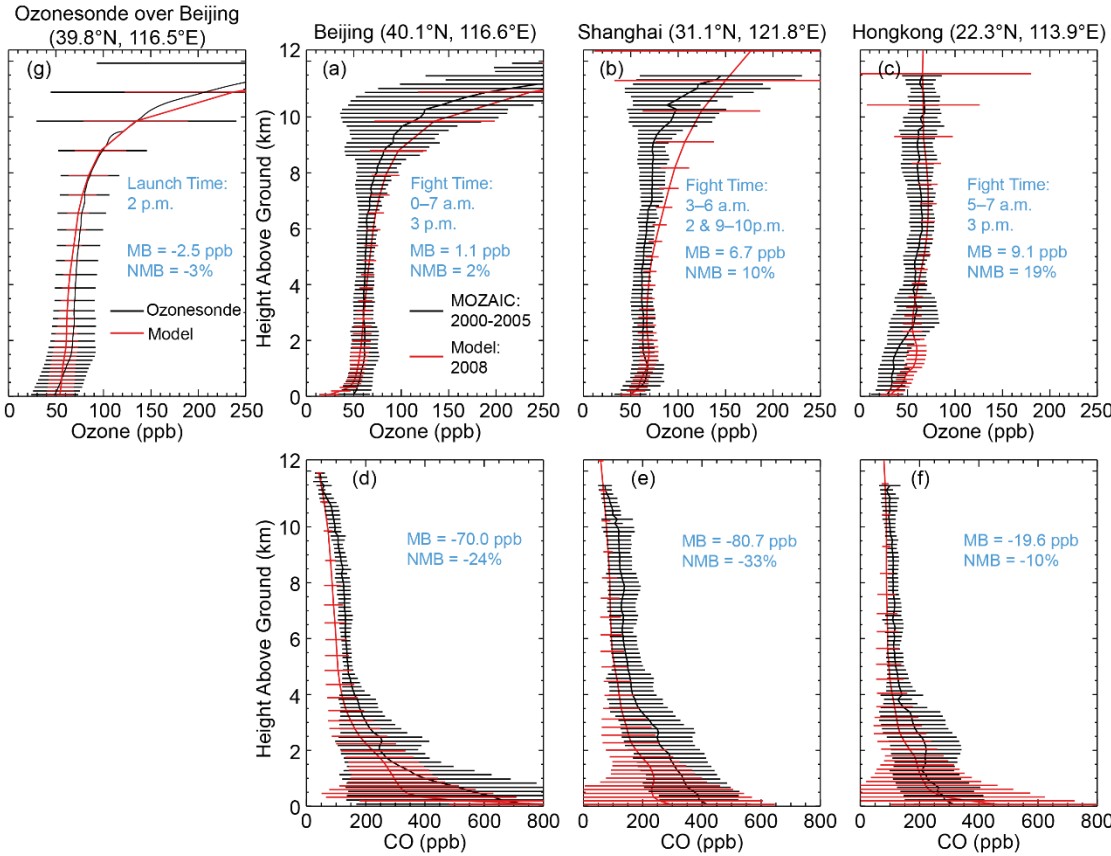


Figure 6. Model and MOZAIC vertical profiles of (a–c) ozone and (d–f) CO over
airports of Beijing, Shanghai and Hong Kong, averaged over multiple profiles. (g)
Model and GPSO3 ozonesonde data over Beijing in spring 2008. Horizontal bars
indicate one standard deviation across multiple profiles. Mean bias (MB), normalized
mean bias (NMB), main fight times (local time) at each MOZAIC site and GPSO3
ozonesonde launch time (local time) are also shown.






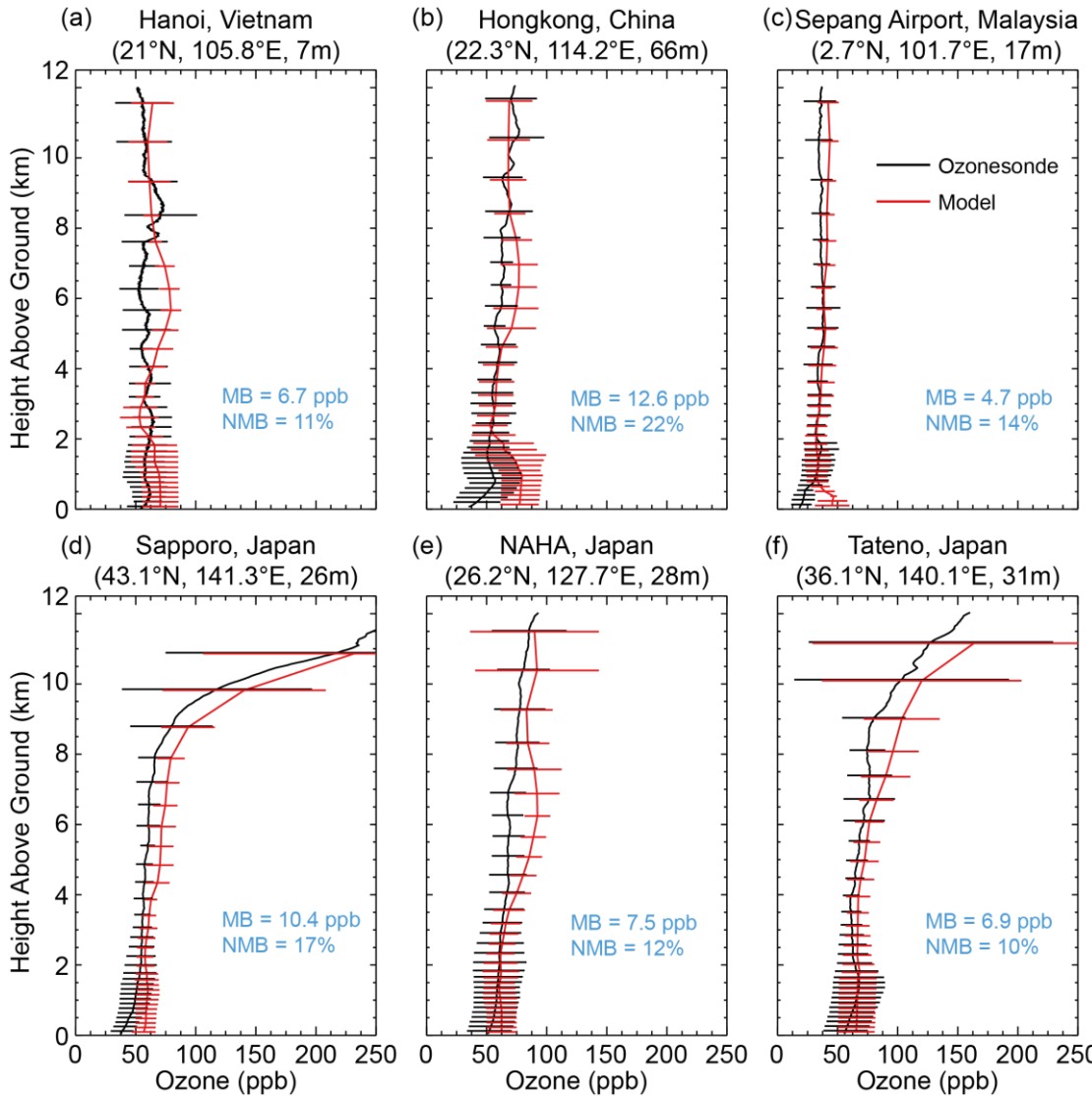


Figure 7. Model and WOUDC ozone profiles at six sites, averaged over multiple
profiles. Horizontal lines indicate one standard deviation across multiple profiles.
Mean bias (MB) and normalized mean bias (NMB) are shown in blue.







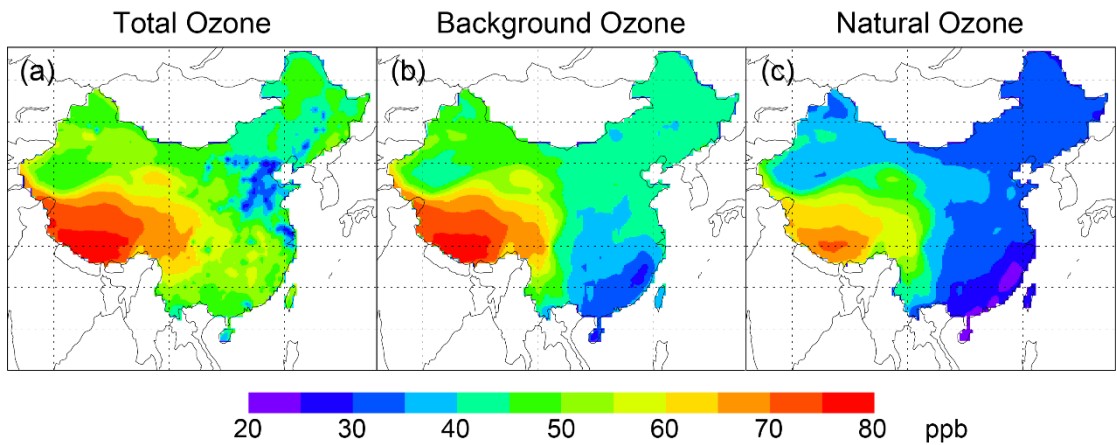


Figure 8. Spatial distribution of springtime daily mean (a) total surface ozone, (b)
background ozone and (c) natural ozone over China.
















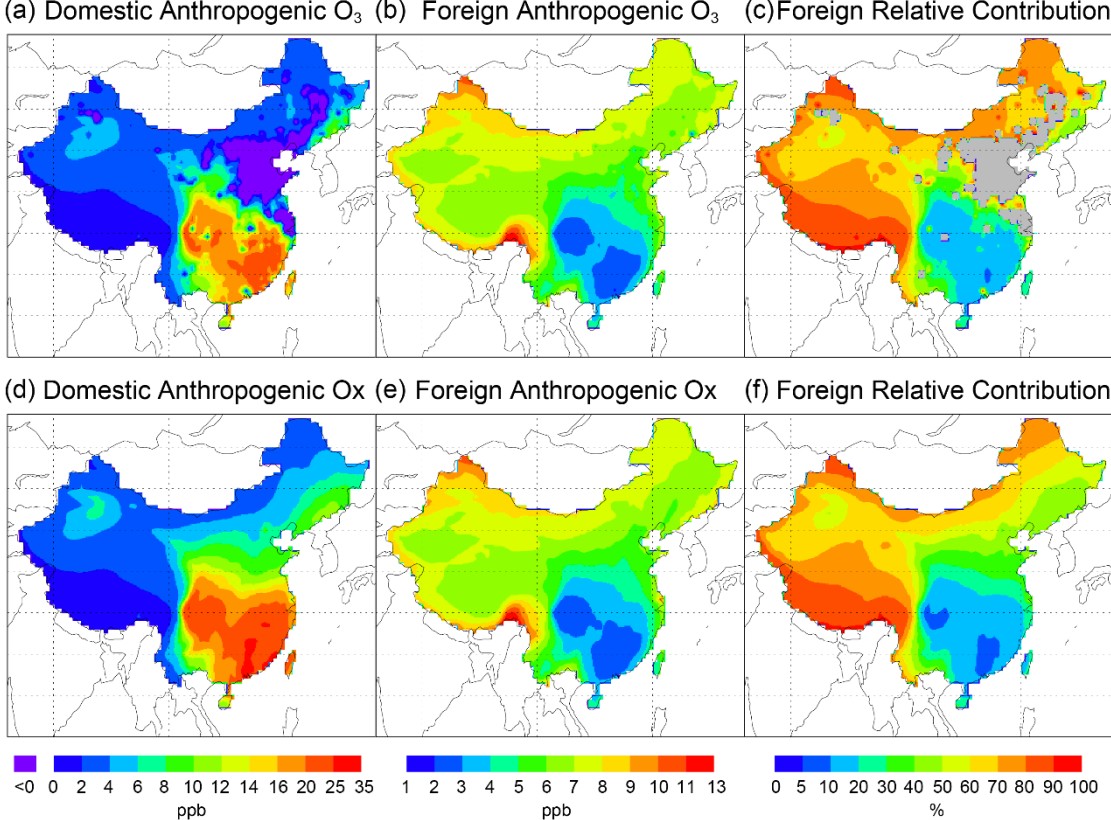


Figure 9. Spatial distribution of springtime daily mean surface ozone over China contributed by (a) domestic and (b) foreign anthropogenic emissions. (c) Percentage contribution of foreign anthropogenic emissions to total anthropogenic ozone; areas with negative Chinese contributions (due to NOx titration) are marked in grey. (d–f) similar to (a–c) but for Ox (= $O_3$ + $NO_2$). The linear weighting adjustment is applied to derive all results. Note that the color scales are different between (a, d) and (b, e).





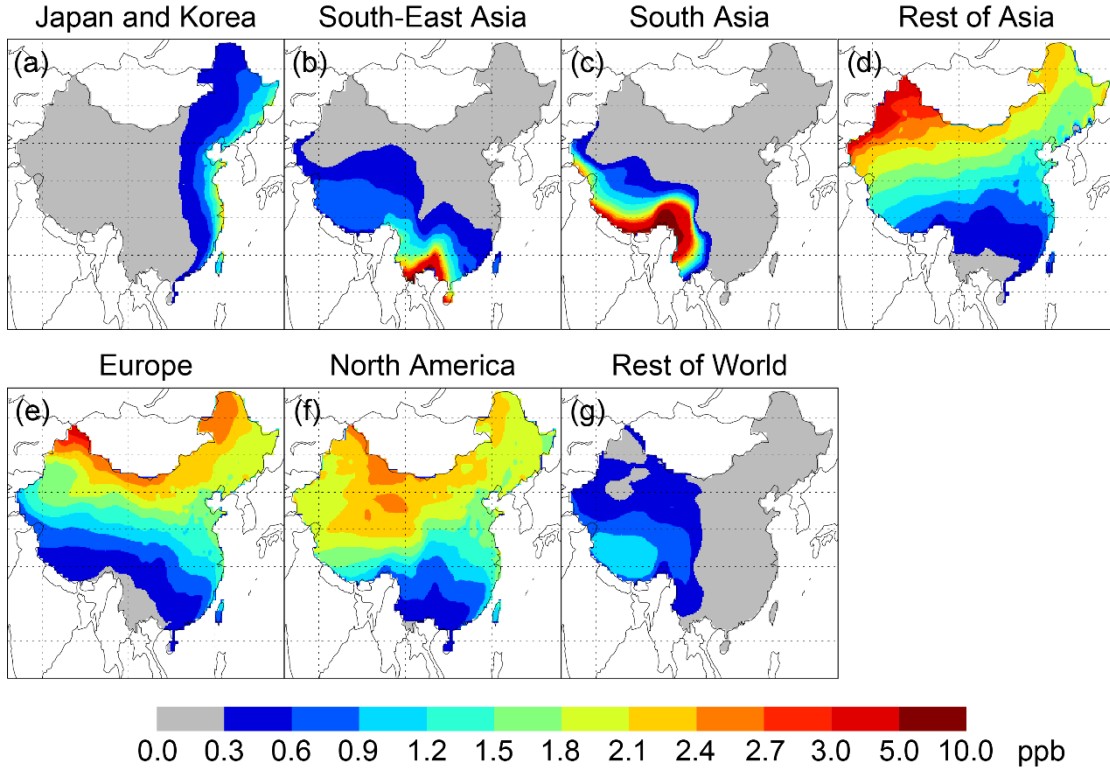

Figure 10. Spatial distribution of springtime daily mean surface ozone over China contributed by anthropogenic emissions of individual regions. The ozone enhancement over China by anthropogenic emissions of each region is determined by difference between the base case simulation CTL and zero-out simulation without that region's anthropogenic emissions, followed by the linear weighting adjustment.

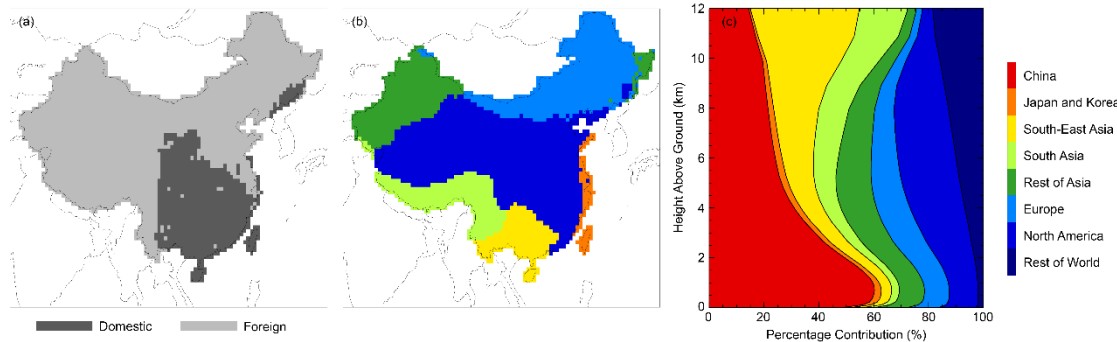


Figure 11. (a) Indication of the largest anthropogenic contributor (domestic or
foreign) to surface ozone at individual locations of China. (b) Indication of the largest
foreign anthropogenic contributor to surface ozone at individual locations of China.
(c) Vertical distribution of percentage contribution of each region to total
anthropogenic ozone over China.











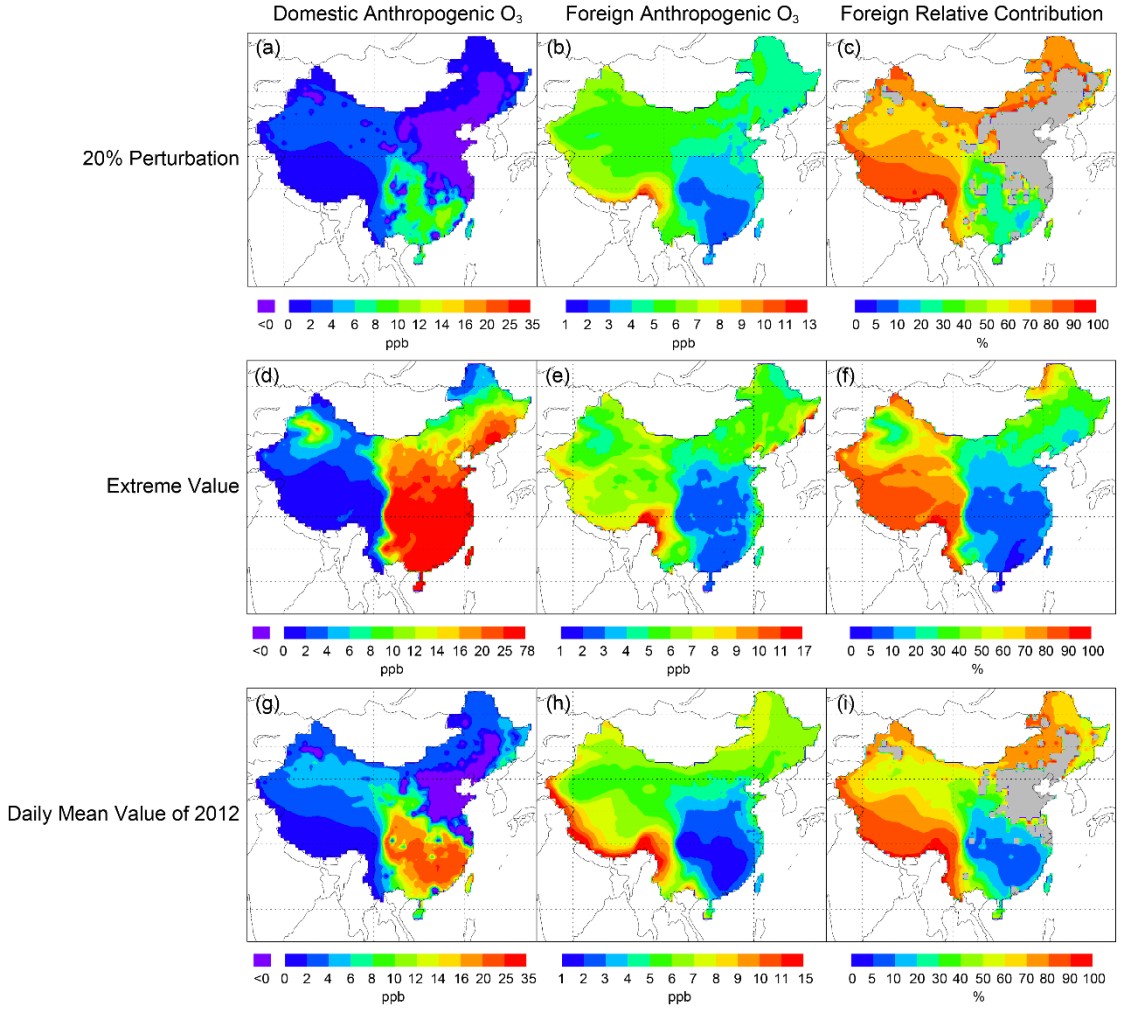

Figure 12. (a–c) similar to Fig. 9a–c but for springtime daily mean ozone calculated by 20% perturbation method. (d–f) similar to Fig. 9a–c but for springtime extreme ozone value (defined as the average of the top 5% hourly ozone concentrations). (g–i) similar to Fig. 9a–c but for springtime daily mean ozone in 2012. The linear weighting adjustment is applied to derive all results. Note that the color scales are different in each panel.

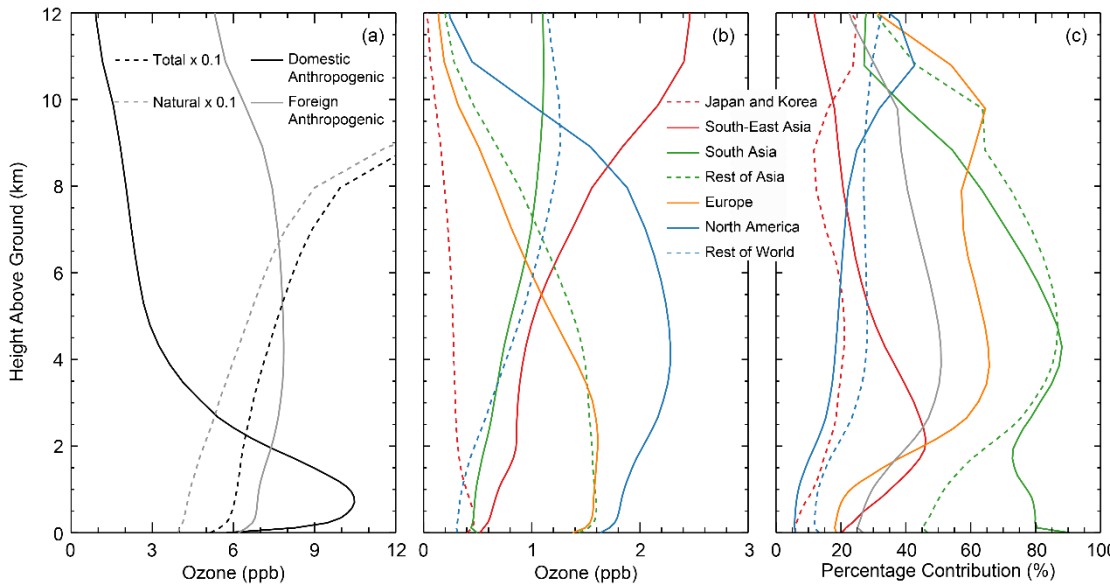

1170

Figure 13. (a) Vertical distribution of China average daily mean ozone contributed by
domestic anthropogenic emissions, foreign anthropogenic emissions, natural sources
(scaled by 0.1) and total sources (scaled by 0.1). (b) Contribution by anthropogenic
emissions of each foreign source region. (c) Of the ozone over China due to
anthropogenic emissions of each foreign region, the portion produced within each
foreign source region's territory calculated based on a combination of zero-out and
tagged simulations. The linear weighting adjustment is applied to derive all results.

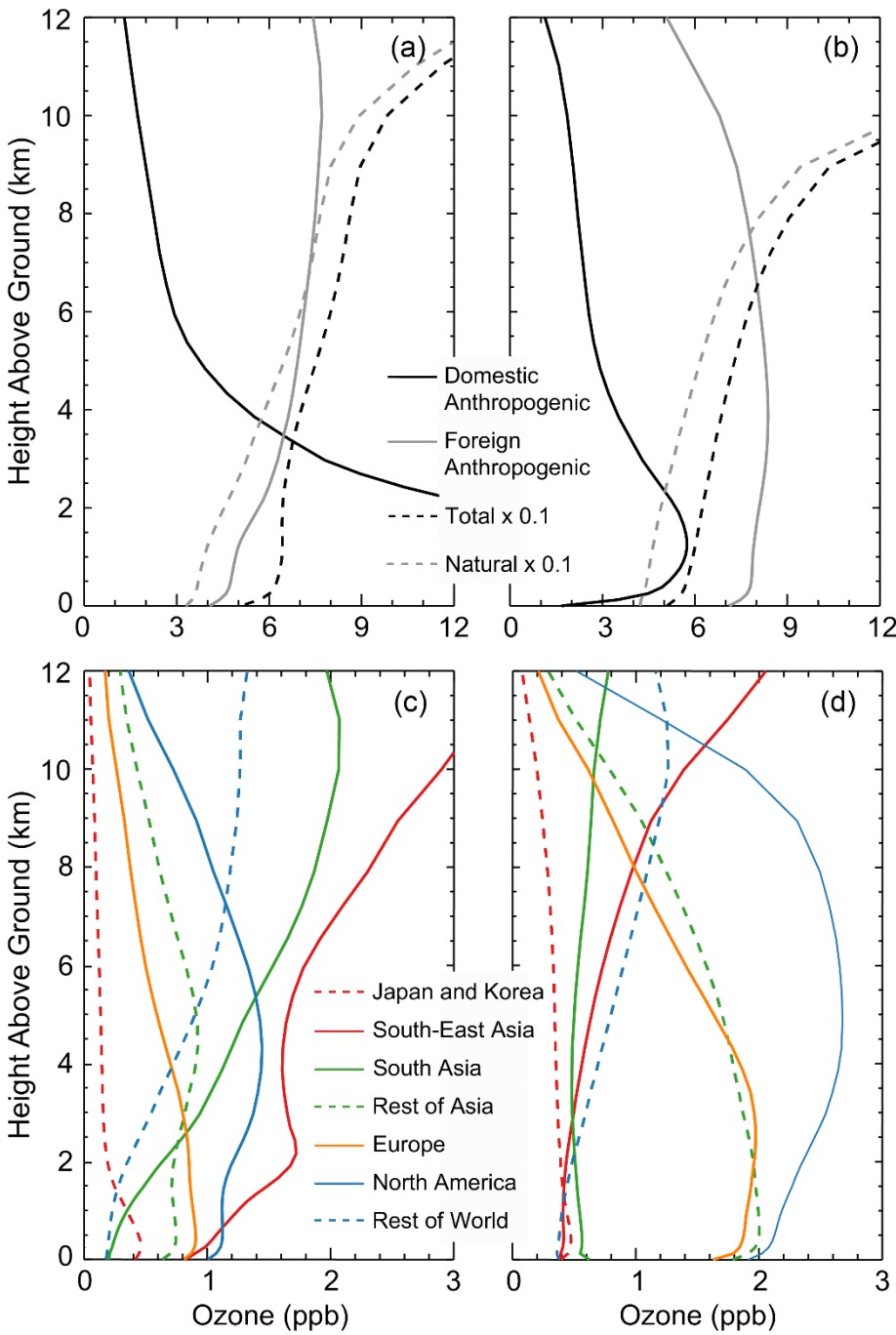


Figure 14. (a) Vertical distribution of regional average daily mean ozone contributed
by domestic anthropogenic emissions, foreign anthropogenic emissions, natural
sources (scaled by 0.1) and total sources (scaled by 0.1) over regions where Chinese
anthropogenic emissions contribute more surface ozone than total foreign
anthropogenic emissions. (c) Contribution by anthropogenic emissions of each foreign
source region over regions where Chinese anthropogenic emissions contribute more
surface ozone than total foreign anthropogenic emissions. (b, d) similar to (a, c) but
for regional average daily mean ozone over regions where foreign anthropogenic
emissions dominate. The linear weighting adjustment is applied to derive all results.