# Peer review of "Foreign and domestic contributions to springtime ozone over China"

_Atmospheric Chemistry and Physics, 2017_

## Referee Comment (RC1) · Anonymous Referee #1 · 13 Feb 2018

General Comments:

This study examines the domestic and foreign influence of anthropogenic emissions on ozone over China using the GEOS-Chem model and two methods of identifying contributions, a "zero-out" approach and a tagging approach (which seems to be missing from the manuscript). After first validating the model's capabilities against surface and ozonesonde observations, they proceed to characterize the spatial influence (horizontally and vertically) of natural, background, foreign anthropogenic and domestic anthropogenic emissions on ozone over China.

Much of the analysis in this manuscript contains significant insights into the ozone chemistry over China and the impact of foreign and domestic emissions on tropospheric ozone. This manuscript could be a valuable contribution to ACP and to our

understanding of ozone attribution over China, but there are several major items that need to be addressed before I can recommend publication. I discuss two major issues below, and conclude with technical comments.

Specific Comments:

First, this study examines a single 3-month period (spring) in 2008 and draws extensive conclusions based on this period. The nature of emissions, ozone chemistry, meteorology, and atmospheric transport make it difficult to believe in the robustness of results drawn from such a short period without some characterization of the trends, variability, and uniqueness/non-uniqueness of this particular spring in 2008. While the authors point out the reasons for selecting this time period (L102-107), and while they mention some of these issues (e.g. NOx trends in L282-285, differences in emissions and meteorology in L311-312), I do not believe there is a sufficient demonstration of the robustness of their results, and there are many questions that need to be addressed. Are the results drawn throughout Sections 4 and 5 robust for different years, or are they sensitive to chemical and meteorological variability and thereby vary from year-to-year? How much to they vary? Where does the spring of 2008 fit into the bigger ozone/chemistry/meteorology context over China?

I feel that either: (1) additional simulations including at least one additional year are required to demonstrate the simulated variability of ozone over China and the robustness of these results; or (2) the manuscript requires additional literature reviews and a careful description of the ozone variability over China as a demonstration of the robustness of the results. In L247-253 the authors discuss an additional year of simulation, which could certainly provide some of this temporal variability context. Some of the publications below could provide some of this context and reasons why 3-months is not long enough to draw strong conclusions, especially with regards to ozone:

Xu, X., Lin, W., Wang, T., Yan, P., Tang, J., Meng, Z., and Wang, Y.: Long-term trend of surface ozone at a regional background station in eastern China 1991–2006: enhanced

variability, Atmos. Chem. Phys., 8, 2595-2607, https://doi.org/10.5194/acp-8-2595-2008, 2008.

Jin, X., and T. Holloway, Spatial and temporal variability of ozone sensitivity over China observed from the Ozone Monitoring Instrument, J. Geophys. Res. Atmos., 120, 7229–7246, doi:10.1002/2015JD023250, 2015

W.N. Wang, T.H. Cheng, X.F. Gu, H. Chen, H. Guo, Y. Wang, F.W. Bao, S.Y. Shi, B.R. Xu, X. Zuo, C. Meng, X.C. Zhang, Assessing spatial and temporal patterns of observed ground-level ozone in China, Sci. Rep., 7 (1), p. 3651, 10.1038/s41598-017-03929-w, 2017.

Garcia-Menendez, F., Monier, E., and Selin, N. E.:The role of natural variability in projections of climate change impacts on U.S. ozone pollution, Geophys. Res. Lett., 44, 2911–2921, 2017.

Brown-Steiner, B., Selin, N. E., Prinn, R. G., Monier, E., Tilmes, S., Emmons, L., and Garcia-Menendez, F.: Maximizing Ozone Signals Among Chemical, Meteorological, and Climatological Variability, Atmos. Chem. Phys. Discuss., https://doi.org/10.5194/acp-2017-954, in review, 2017.

Second, the manuscript at times leaves out critical information or does not sufficiently describe methods, definitions, and figures. While this manuscript contains many valuable results, there were several moments where I didn't feel there was enough information provided to understand what was done, or why it was done, and times when I had to search for descriptions and/or infer some explanations on my own. The following list summarizes areas and issues that need to be addressed and revised:

(1) The authors state that they combine zero-out simulations with tagged ozone simulations (and the tagged ozone simulations are mentioned in Table 2 and on L110, L114, and L183-187), but nowhere throughout the rest of the manuscript are the tagged ozone results described or shown. Were they not used? Where are the descriptions of

these results?

(2) A linear weighting method is used to adjust the ozone attribution results, and is described on L188-195, but the description is insufficient. I am not familiar with this method, so I do not fully understand what Equation 1 means, and tracing back to the Li et al. (2016a) citation brings me to a 'normalized marginal method' used for radiative forcing attribution, not ozone attribution. It is not clear to me where the precise formulation of Equation 1 came from, what it does, what impact the adjustment has on the results, or why it was selected.

(3) Many of the comparisons to observations compared the simulated spring of 2008 with other years (e.g. L281-285, L297-298, L311-312), and given the variability in ozone, chemistry, and meteorology (see above), I'm not sure these are wholly valid comparisons, especially without the broader temporal context of ozone over China. Some sort of quantification of measurement-model uncertainty and sensitivity to the time periods compared needs to be included.

(4) The authors define 'natural ozone' on L353, 'background ozone' on L363, 'domestic anthropogenic ozone' on L369, but do not define 'foreign anthropogenic ozone,' leaving it to the reader to infer a definition. Also, I'm not sure that 'natural ozone' is an accurate description of what is described, as humans have influenced atmospheric chemistry beyond just anthropogenic emissions, perhaps 'non-anthropogenic ozone' instead?

(5) Figure 10 should include a plot of the regions where Chinese emissions are the dominant contributor. Figure 10b shows that on average, China contributes ∼50% to surface ozone, and it's clear from Figure 8 that Chinese emissions dominate southeastern China's ozone. I'm not sure it's worthwhile then to point out the dominant foreign contribution to surface ozone over regions where Chinese emissions are dominant, especially when the foreign contribution is so low (Figure 8b,c). On its own, Figure 1a is an incomplete representation.

(6) Figure 11 is hard to parse, and given the large spatial heterogeneity shown in the

other Figures, it is not clear to me that a single vertical plot averaging all of China provides valuable information, or if it muddles interesting information through the averaging. This also applies to Figure 10b. Perhaps split these vertical profiles up into regions dominated by domestic and foreign contributions? Or perhaps apply some population weighing? In addition, Figure 11a should also include total ozone and a comparison should be made of total ozone (from the CTL run) and the sum of natural ozone, domestic anthropogenic ozone, and foreign anthropogenic ozone. It's not clear that these will match up, but it would speak to the non-linearity of the ozone simulations and contribution sensitivity simulations. Finally, I had difficulty in understanding Figure 11c as I initially assumed that Figure 11c was just a reformulation of Figure 11b in percentages rather than ppbv. The caption of the figure and the description on L484-485 are not clear, and as there is no description of how they arrived at this calculation, I'm unsure precisely what Figure 11c plots. The analysis summarized in these plots is interesting, but as is I have more questions that could be answered by subdividing these plots.

Technical Corrections:

Throughout the manuscript there are many acronyms that are used but not defined (e.g. MOZART, NAQPMS, PKUCPL).

L91: The ozone itself doesn't differ, but the plumes and chemical regimes, which produce and destroy the ozone, does differ.

L156-171: This paragraph mostly duplicates the information already in Table 1, and I do not feel that this redundancy is necessary.

L198-199: Figure 3 should be Figure 2

L250: These numbers do not match those found in Table 3

L265-266: The authors claim that the biases are due to overestimated free tropospheric and stratospheric transport, but it's not clear to me how this conclusion was reached.

The color scales in Figures 8a,b,d,e need to be consistent, as it requires extra effort to compare the Chinese Anthropogenic and Foreign Anthropogenic contributions. There is a risk that a casual reader would assume that the color scales in Figures 8a,b,d,e are the same, which would lead to incorrect conclusions.

L384: I don't feel that describing the air over the Sichuan Basin as "more isolated" is the correct description; rather the ozone chemistry of the region is controlled and dominated by domestic emissions and chemistry rather than foreign emissions.

---

## Referee Comment (RC2) · Anonymous Referee #2 · 24 Feb 2018

The manuscript presents a modeling analysis and attributes ozone in China to anthropogenic emissions outside China. Basically, it represents a breakdown of background ozone in China to different foreign regions. Although this breakdown analysis has its value, my main concerns are (1) the analysis was limited to the seasonal mean ozone attribution rather than high ozone events, (2) the modeling was based on a single, non-recent, year (2008), and (3) the nonlinearity in source attribution seems to be large and needs to be assessed more carefully. These issues need to be addressed and corrected before this work can be accepted by ACP.

Major comments

1. The last sentence of the abstract, "Global emission reduction is critical for China's ozone mitigation", should be removed. The reported contribution of foreign emissions

on ozone in China is essentially the background ozone. It has been well established (e.g. by several HTAP reports and references therein) that background ozone is substantial (20-50 ppbv) everywhere in the northern mid-latitude continents. For long-lived air pollutants such as ozone, essentially every country pollutes others and vice versa. To effectively mitigate ozone pollution in China, the key is to understand which source region drives the variability, especially of the high ozone days. I would be surprised if the foreign contribution is a primary factor for day-to-day changes of peak ozone over the majority of China. It appears that the paper only focuses on the seasonal mean contributions from foreign sources, thus the last sentence is a premature statement and may be interpreted misleadingly that domestic emissions control is not important.

2. To follow up the previous comment, by reporting just seasonal mean contributions of foreign sources on ozone in China, I feel the paper does not add much new knowledge to the field, especially considering their analysis was based on a single year's simulation (see my next comment). The paper would be interesting if they had analyzed the foreign contribution to peak ozone events (during pollution episode) in addition to the mean ozone.

3. I have concerns about the choice of a single, non-recent, year (2008) used in the paper for the whole analysis. The exact magnitudes of ozone mixing ratio attributable to different sources depend on meteorology and emissions, both linked with the year of simulation. How would these ozone values change if another year is chosen to conduct the analysis? The authors stated that routine ozone measurements were scarce before 2013 (pg 6, line 203), so why not a simulation year after 2013? This would be more desirable to take advantage of more observational data for model evaluation. In particular, the increase of ozone pollution is a more recent concern in Chinese cities, after high PM events are on the decline.

4. I am also concerned with the statement that over the polluted eastern China, "Chinese anthropogenic emissions lead to reductions (instead of enhancements) of surface ozone" (pg 10, line 374-375). The authors attributed this to the ozone titration effect

by freshly emitted NO. The phenomena do occur in urban areas, but the GEOS-Chem simulation used in this study has a relatively coarse grid cell even for the nested-grid option (∼50km x 50 km). This resolution would substantially smear out NOx emissions in a grid, leading to muted titration effect. My interpretation of that statement is that it suggests the nonlinearity in the zero-out simulations is strong (because it leads to negative ozone changes) and needs to be tested via different sensitivity runs and dealt with carefully. For example, the authors could try zeroing-out foreign anthropogenic emissions instead of Chinese anthropogenic emissions or try reducing Chinese emissions by a certain percentage rather than a complete zero-out, and then analyze if the different perturbation runs give consistent results over North China.

Minor Issues

Pg 5, line 190-195: The description of the weighting method to account for nonlinear chemistry is very vague, and I don't understand the scientific basis for this method. It should be expanded and explained in a way such that it is understandable to readers who have not read the original Li et al (2016) paper.

Pg 3, line 108-109: this sentence is confusing. Do you mean there are 10 producing regions and 8 source regions? Why and how are they different?

Language Issues: The paper has a few grammar errors and language issues, some examples listed below. I would suggest the authors proofread it more carefully during the revision stage.

Pg 1, line 6: "mean bias at 10-15%" should be "mean bias of 10-15%"

Pg 2, line 38: "at surface" should be "at the surface".

Pg 10, line 356: "nature ozone" should be "natural ozone".

---

## Author Comment (AC1) · 14 Jun 2018

**Reviewer 1**

General Comments:

This study examines the domestic and foreign influence of anthropogenic emissions on ozone over China using the GEOS-Chem model and two methods of identifying contributions, a "zero-out" approach and a tagging approach (which seems to be missing from the manuscript). After first validating the model's capabilities against surface and ozonesonde observations, they proceed to characterize the spatial influence (horizontally and vertically) of natural, background, foreign anthropogenic and domestic anthropogenic emissions on ozone over China.

Much of the analysis in this manuscript contains significant insights into the ozone chemistry over China and the impact of foreign and domestic emissions on tropospheric ozone. This manuscript could be a valuable contribution to ACP and to our understanding of ozone attribution over China, but there are several major items that need to be addressed before I can recommend publication. I discuss two major issues below, and conclude with technical comments.

We thank the referee for helpful comments. We respond to each comment below. The referee comments are shown in red. Our replies are shown in black.

Specific Comments:

First, this study examines a single 3-month period (spring) in 2008 and draws extensive conclusions based on this period. The nature of emissions, ozone chemistry, meteorology, and atmospheric transport make it difficult to believe in the robustness of results drawn from such a short period without some characterization of the trends, variability, and uniqueness/non-uniqueness of this particular spring in 2008. While the authors point out the reasons for selecting this time period (L102-107), and while they mention some of these issues (e.g. NOx trends in L282-285, differences in emissions and meteorology in L311-312), I do not believe there is a sufficient demonstration of the robustness of their results, and there are many questions that need to be addressed. Are the results drawn throughout Sections 4 and 5 robust for different years, or are they sensitive to chemical and meteorological variability and thereby vary from year-to-year? How much to they vary? Where does the spring of 2008 fit into the bigger ozone/chemistry/meteorology context over China?

I feel that either: (1) additional simulations including at least one additional year are required to demonstrate the simulated variability of ozone over China and the robustness of these results; or (2) the manuscript requires additional literature reviews and a careful description of the ozone variability over China as a demonstration of the robustness of the results. In L247-253 the authors discuss an additional year of simulation, which could certainly provide some of this temporal variability context. Some of the publications below could provide some of this context and reasons why 3-months is not long enough to draw strong conclusions, especially with regards to ozone:

Xu, X., Lin, W., Wang, T., Yan, P., Tang, J., Meng, Z., and Wang, Y.: Long-term trend of surface ozone at a regional background station in eastern China 1991–2006: enhanced

variability, Atmos. Chem. Phys., 8, 2595-2607, https://doi.org/10.5194/acp-8-2595-2008, 2008.

Jin, X., and T. Holloway, Spatial and temporal variability of ozone sensitivity over China observed from the Ozone Monitoring Instrument, J. Geophys. Res. Atmos., 120, 7229–7246, doi:10.1002/2015JD023250, 2015

W.N. Wang, T.H. Cheng, X.F. Gu, H. Chen, H. Guo, Y. Wang, F.W. Bao, S.Y. Shi, B.R. Xu, X. Zuo, C. Meng, X.C. Zhang, Assessing spatial and temporal patterns of observed ground-level ozone in China, Sci. Rep., 7 (1), p. 3651, 10.1038/s41598-017-03929-w, 2017.

Garcia-Menendez, F., Monier, E., and Selin, N. E.:The role of natural variability in projections of climate change impacts on U.S. ozone pollution, Geophys. Res. Lett., 44, 2911–2921, 2017.

Brown-Steiner, B., Selin, N. E., Prinn, R. G., Monier, E., Tilmes, S., Emmons, L., and Garcia-Menendez, F.: Maximizing Ozone Signals Among Chemical, Meteorological, and Climatological Variability, Atmos. Chem. Phys. Discuss., https://doi.org/10.5194/acp-2017-954, in review, 2017.

Thanks for your suggestion.

As in the newly added Sect. 4.3 (Line 492-516), previous studies have shown notable interannual variability in surface ozone over China driven by changes in precursor emissions and meteorology (Xu et al., 2008; Jin et al., 2015; Wang et al., 2017). To test how the interannual variability of meteorology and emissions would affect our source attribution findings, we have repeated all zero-out runs for spring 2012, the latest year when the GEOS-5 meteorological fields are available. Emissions for 2012 were adopted from the Community Emissions Data System (CEDS) inventory (Hoesly et al., 2018); 2012 is also the latest year the CEDS emissions for China are adjusted by the MEIC inventory. Table A1 shows the anthropogenic emissions in the two years. All zero-out simulation results in 2012 underwent the same linear weighting adjustment as for those in 2008. Figure A1d–f show the results for domestic versus foreign contributed ozone in spring 2012, as compared to the results for spring 2008 (adopted from Fig. 9a–c in the revised paper). In absolute terms, Chinese contributed ozone are similar between 2008 and 2012 (comparing Fig. A1a and d), reflecting the slight changes in domestic precursor emissions (Table A1). From 2008 to 2012, the absolute foreign contributed ozone increase along the southern boarder due to much enhanced emissions in South-East Asia and South Asia. The absolute foreign contributions decrease over the north and south, reflecting the net effect of changes in European and North American emissions (within 20% for both NOx and NMVOC), increased emissions in Rest of Asia, and changes in meteorology. In relative terms (Fig. A1c and f), the percentage foreign anthropogenic contributions to total anthropogenic ozone decrease from 2008 to 2012 over southern China. Nonetheless, in both years the percentage foreign contributions exceed 50% over western China and are 5–40% over southern China. Therefore our general finding that both foreign and domestic contributions to Chinese anthropogenic ozone are important holds true for these two years.

[Figure]

Figure A1. Spatial distribution of springtime daily mean surface ozone over China contributed by (a) domestic and (b) foreign anthropogenic emissions in 2008. (c) Percentage contribution of foreign anthropogenic emissions to total anthropogenic ozone in 2008; areas with negative Chinese contributions (due to NOx titration) are marked in grey. (d–f) similar to (a–c) but for results of 2012. The linear weighting adjustment is applied to derive all results. Please note that the color scales are different between (a, d) and (b, e).

Table A1. Springtime anthropogenic emissions of NOx, CO and NMVOC in 2008 and 2012 in each source region defined in Fig. 1.

| 2008 | China | Japan and Korea | South-East Asia | South Asia | Rest of Asia | Europe | North America | Rest of world |
|---|---|---|---|---|---|---|---|---|
| NOx (TgN) | 2.0 | 0.3 | 0.4 | 0.4 | 0.7 | 1.2 | 1.3 | 1.0 |
| CO (Tg) | 42.3 | 1.7 | 10.9 | 16.7 | 10.0 | 12.5 | 17.7 | 25.5 |
| NMVOC (TgC) | 2.9 | 0.2 | 1.3 | 1.3 | 1.1 | 1.1 | 2.1 | 1.9 |
| 2012 | | | | | | | | |
| NOx (TgN) | 2.2 | 0.3 | 0.6 | 1.3 | 1.0 | 1.0 | 1.1 | 1.5 |
| CO (Tg) | 39.2 | 2.4 | 15.4 | 21.3 | 8.9 | 7.9 | 13.1 | 38.0 |
| NMVOC (TgC) | 3.0 | 0.2 | 3.0 | 2.4 | 2.3 | 1.2 | 1.8 | 6.8 |

Reference:

Hoesly, R. M., Smith, S. J., Feng, L., Klimont, Z., Janssens-Maenhout, G., Pitkanen, T., Seibert, J. J., Vu, L., Andres, R. J., Bolt, R. M., Bond, T. C., Dawidowski, L., Kholod, N., Kurokawa, J.-I., Li, M., Liu, L., Lu, Z., Moura, M. C. P., O'Rourke, P. R., and Zhang, Q.: Historical (1750–2014) anthropogenic emissions of reactive gases and aerosols from the Community Emissions Data System (CEDS), Geosci. Model Dev., 11, 369-408, https://doi.org/10.5194/gmd-11-369-2018, 2018

Second, the manuscript at times leaves out critical information or does not sufficiently describe methods, definitions, and figures. While this manuscript contains many valuable results, there were several moments where I didn't feel there was enough information provided to understand what was done, or why it was done, and times when I had to search for descriptions and/or infer some explanations on my own. The following list summarizes areas and issues that need to be addressed and revised:

(1) The authors state that they combine zero-out simulations with tagged ozone simulations (and the tagged ozone simulations are mentioned in Table 2 and on L110, L114, and L183-187), but nowhere throughout the rest of the manuscript are the tagged ozone results described or shown. Were they not used? Where are the descriptions of these results?

As mentioned in our original manuscript (Line 84–85), ozone over China attributed to anthropogenic emissions of an emission source region can be produced both within the domain of that source region and outside the domain due to the outflow of ozone precursors. The zero-out simulations provide the total transboundary ozone due to emissions of a source region. The tagged ozone approach quantifies the ozone produced in any designated region, with no information about whether the associated precursors are emitted in that region or are transported from somewhere else.

We combined tagged ozone simulations with the zero-out method to quantify the contribution of ozone over China attributed to anthropogenic emissions of each source region produced *within* and *outside* that source region, respectively. Results combining

tagged ozone simulations and zero-out simulations are shown in Sect. 5 and Fig. 13c (Fig. 11c in the original manuscript).

For further explanation, here we take ozone over China attributed to European anthropogenic emissions as an example. The tagged ozone approach quantifies ozone over China produced in any designated region (due to global emissions), which is defined as an artificial "tracer" in the tagged simulation. We defined 10 producing regions and thus 10 artificial tracers in tagged simulations, including eight tropospheric above-land domains (China, Europe, etc.), tropospheric above-ocean domain, and the stratosphere. To complement the full-chemistry control case (CTL), we ran the tagged simulation to calculate the contributions from these 10 producing domains (and 10 artificial tracers) (T_CTL). For the zero-European-anthropogenic-emissions case (xEU, a zero-out simulation), we did a similar calculation (T_xEU). Thus, the difference between CTL and xEU gave the total ozone due to European anthropogenic emissions, and the difference between T_CTL and T_xEU gave the concentration of ozone produced over each of these 10 producing domains due to European anthropogenic emissions. To account for the effect of chemical nonlinearity in these attribution analyses, we further applied a weighting to these results.

We have revised the introduction of the new Fig. 13c (the old Fig. 11c) (Line 538-542) as "Figure 13c further separates the portion of ozone produced within each source region's territory from the portion produced outside of that source region; results here were derived from a combination of zero-out simulations (e.g., CTL and xEU) and tagged simulations (e.g., T_CTL and T_xEU)."

(2) A linear weighting method is used to adjust the ozone attribution results, and is described on L188-195, but the description is insufficient. I am not familiar with this method, so I do not fully understand what Equation 1 means, and tracing back to the Li et al. (2016a) citation brings me to a 'normalized marginal method' used for radiative forcing attribution, not ozone attribution. It is not clear to me where the precise formulation of Equation 1 came from, what it does, what impact the adjustment has on the results, or why it was selected.

Ozone production is nonlinearly dependent on its precursors. Thus, the sum of natural ozone and anthropogenic ozone due to each emission source region calculated from zero-out simulations is not equal to ozone concentration calculated in the control run (CTL). Considering uncertainties induced by emission perturbation methods, we used a linear weighting method to adjust ozone concentration attributed to different sources, ensuring that the sum of natural ozone and anthropogenic ozone in zero-out simulations is equal to amount of ozone simulated in CTL.

As clarified in the revised manuscript, here is an example to adjust Chinese contribution to ozone over China using the linear weighting approach. Equation A1 calculates the fractional Chinese contribution ($\alpha$) to the sum of ozone from individual anthropogenic source regions and from natural sources; the simulations involved are all full-chemistry runs (CTL, xCH, xEU, …, xANTH). Equation A2 applies the fractional contribution $\alpha$ to the total ozone in CTL to obtain the final adjusted Chinese contribution. These equations are used in the revised manuscript for better clarity; they are simply a transformed version of Eq. 1 in the original manuscript.

Similar adjustments were applied to other source regions, such that all results shown in our original manuscript are for "adjusted" ozone attribution through this linear weighting approach.

As shown in our revised manuscript Line 216–218, "A similar approach was used by Li et al. (2016a) to estimate the contribution of China to global radiative forcing, although in their study 20% (instead of 100%) of emissions over individual emission source regions

are removed in the sensitivity simulations."

$$\alpha = \frac{Con(CTL)-Con(xCH)}{\sum_{i=1}^{8}[Con(CTL)-Con(Ci)]+Con(xANTH)} \tag{A1}$$

$$C_{CH} = \alpha \times Con(CTL) = \frac{Con(CTL)-Con(xCH)}{\sum_{i=1}^{8}[Con(CTL)-Con(Ci)]+Con(xANTH)} \times Con(CTL) \tag{A2}$$

(3) Many of the comparisons to observations compared the simulated spring of 2008 with other years (e.g. L281-285, L297-298, L311-312), and given the variability in ozone, chemistry, and meteorology (see above), I'm not sure these are wholly valid comparisons, especially without the broader temporal context of ozone over China. Some sort of quantification of measurement-model uncertainty and sensitivity to the time periods compared needs to be included.

In our study, in order to use as many observations to constrain model ozone as possible, we included a suite of measurement data in spring 2008 and in other years. For surface ozone, we focused on the comparison with observations in 2008 that are temporally consistent with our simulation; we showed the day-to-day variation at those sites. We extended the comparison to surface measurements in other years, in order to give a sense of how model ozone is situated in the general ozone pollution phenomena, as also explained in the revised manuscript. For vertical profiles, we have tried our best to match the time of observations and model simulations. For comparison with MOZAIC from earlier years, we are more concerned with the general vertical shape, given the trends and interannual variability. Long-term observations indicate strong ozone growth over China due to changes in domestic precursor emissions (e.g., Wang et al., 2009; Xia et al., 2016). This growth is consistent with our results that model ozone in 2008 are generally higher than observations in earlier years, although the vertical shape is captured fairly well.

In explaining Table 4, we have revised the text Line 307–315 as follows:

"The model has a large overestimate by 48% at the Hok Tsui coastal rural site in Hong Kong (36.0 versus 53.4 ppb), although the times are different (2008 versus 1994–2007). Wang et al. (2009) shows that the springtime ozone concentration at this site increased from 1994 to 2007 at a rate of 0.41 ppb/yr, partly explaining this difference. The remaining difference may reflect that the model resolution is not able to represent the complex local terrain and land-sea contrast at this site. The model overestimates ozone at an urban site in Nanjing by 16%, although the observations were made in 2000–2002 when Chinese anthropogenic emissions of NOx were only about half of those in 2008 (Xia et al., 2016)."

In explaining the comparison with MOZIC profiles, we revised the text Line 339–342 as follows:

"The model overestimates ozone in the middle and upper troposphere over Shanghai, with larger biases at higher altitudes, likely indicating too strong STE. Other causes may include differences in meteorology and growth in emissions between 2000–2005 and 2008, as discussed for the surface ozone in Sect. 3.1."

(4) The authors define 'natural ozone' on L353, 'background ozone' on L363, 'domestic anthropogenic ozone' on L369, but do not define 'foreign anthropogenic ozone,' leaving

it to the reader to infer a definition. Also, I'm not sure that 'natural ozone' is an accurate description of what is described, as humans have influenced atmospheric chemistry beyond just anthropogenic emissions, perhaps 'non-anthropogenic ozone' instead?

Anthropogenic ozone of each foreign region is defined as the difference between the base simulation CTL and each zero-out simulation with no anthropogenic emissions in that foreign region (e.g., xEU), followed by a linear weighting adjustment to account for chemical nonlinearly (Eq. 1 and 2). The total foreign anthropogenic ozone is determined by adding each foreign region's anthropogenic ozone contribution together.

We agree that human behaviors have also affected the climate and other processes that in turn will affect the chemical environment. We used the term "natural ozone" to be consistent with the literature in this area (e.g., Wang et al., 2011).

(5) Figure 10 should include a plot of the regions where Chinese emissions are the dominant contributor. Figure 10b shows that on average, China contributes ~50% to surface ozone, and it's clear from Figure 8 that Chinese emissions dominate southeastern China's ozone. I'm not sure it's worthwhile then to point out the dominant foreign contribution to surface ozone over regions where Chinese emissions are dominant, especially when the foreign contribution is so low (Figure 8b, c). On its own, Figure 1a is an incomplete representation.

Thanks for your suggestion. The regions where Chinese emissions are the dominant contributor are shown in Figure A2a. We have also added this plot into the new Fig. 11 (old Fig. 10).

We have added in the revised manuscript Line 448–453 that:

"Figure 11a shows whether Chinese or foreign anthropogenic contributions are higher at individual locations. Chinese anthropogenic contributions are higher than foreign contributions over southern China and parts of northern China. However, foreign anthropogenic contributions exceed domestic contributions over western China and most of the north, including the populated North China Plain. Over western China, foreign emissions contribute 70–90% of the total anthropogenic ozone (Fig. 9c)."

[Figure]

Figure A2. (a) Indication of the largest anthropogenic contributor (domestic versus foreign) to surface ozone at individual locations of China. (b) Indication of the largest foreign anthropogenic contributor to surface ozone at individual locations of China. (c) Vertical distribution of percentage contribution of each region to total anthropogenic ozone over China.

(6) Figure 11 is hard to parse, and given the large spatial heterogeneity shown in the other Figures, it is not clear to me that a single vertical plot averaging all of China provides

valuable information, or if it muddles interesting information through the averaging. This also applies to Figure 10b. Perhaps split these vertical profiles up into regions dominated by domestic and foreign contributions? Or perhaps apply some population weighing? In addition, Figure 11a should also include total ozone and a comparison should be made of total ozone (from the CTL run) and the sum of natural ozone, domestic anthropogenic ozone, and foreign anthropogenic ozone. It's not clear that these will match up, but it would speak to the non-linearity of the ozone simulations and contribution sensitivity simulations. Finally, I had difficulty in understanding Figure 11c as I initially assumed that Figure 11c was just a reformulation of Figure 11b in percentages rather than ppbv. The caption of the figure and the description on L484-485 are not clear, and as there is no description of how they arrived at this calculation, I'm unsure precisely what Figure 11c plots. The analysis summarized in these plots is interesting, but as is I have more questions that could be answered by subdividing these plots.

We have added a new figure (Fig. 14, also shown here as Fig. A3) with two sets of plots, one for the average over regions where Chinese anthropogenic emissions contribute more surface ozone than total foreign anthropogenic emissions (i.e., southern China), and the other for the regions where foreign anthropogenic emissions dominate.

As also discussed in the end of revised Sect. 5, even over areas where domestic contributions to near-surface ozone exceed total foreign contributions, the regional average ozone contributed by foreign emissions exceeds those contributed by domestic emissions above 3.5 km (Fig. A3a). Figure A3c and d further shows that the (relative) vertical shape of regional average ozone contributed by each foreign source region is similar to the shape of China averaged results in Fig. 13b, although the absolute values (in ppb) are different.

[Figure]

Figure A3. (a) Vertical distribution of regional average daily mean ozone contributed by domestic anthropogenic emissions, foreign anthropogenic emissions, natural sources (scaled by 0.1) and total sources (scaled by 0.1) over regions where Chinese anthropogenic emissions contribute more surface ozone than total foreign anthropogenic

emissions. (c) Contribution by anthropogenic emissions of each foreign source region over regions where Chinese anthropogenic emissions contribute more surface ozone than total foreign anthropogenic emissions. (b, d) similar to (a, c) but for regional average daily mean ozone over regions where foreign anthropogenic emissions dominate. The linear weighting adjustment is applied to derive all results.

We have added total ozone from CTL into the new Fig. 13a (old Fig. 11a, also shown here as Fig. A4a). We also added a new plot in the revised manuscript (new Fig. 2b, also shown here as Fig. A4b) to compare vertical profile of pre-linear-weighting-adjustment sum of natural ozone, domestic anthropogenic ozone and foreign anthropogenic ozone with China average total ozone from CTL .

In all of our results, the linear weighting method is applied to remove the effect of ozone nonlinearity, therefore the total ozone simulated in CTL is equal to the "adjusted" sum of natural ozone, domestic anthropogenic ozone and foreign anthropogenic ozone.

As shown in the revised Sect. 2.2 Line 206–215,

"Figure 2a shows the spatial distribution of the ratio of total surface ozone in CTL to the pre-linear-weighting-adjustment sum of natural ozone, domestic anthropogenic ozone and foreign anthropogenic ozone. The ratio is close to unity over central and western China. Over most of the eastern regions, the ratio is between 1.05 and 1.10, although it can reach 1.30 at a few locations. Figure 2b further compares the vertical profile of China average total ozone in CTL and the profile of pre-linear-weighting-adjustment sum of natural ozone, domestic anthropogenic ozone and foreign anthropogenic ozone. The difference between the two profiles is rather small. These results suggest relative small effects of chemical nonlinearity. And the linear weighting adjustment further removes these effects."

We have revised the caption of old Fig. 11c (new Fig. 13c) as: "Of the ozone over China due to anthropogenic emissions of each foreign region, the portion produced within each foreign source region's territory calculated based on a combination of zero-out and tagged simulations." Please see our further explanations of the use of zero-out and tagged simulations in the response to Q1 and Q2 above.

[Figure]

Figure A4. (a) Vertical distribution of China average daily mean ozone contributed by domestic anthropogenic emissions, foreign anthropogenic emissions, natural sources (scaled by 0.1) and total source (scaled by 0.1). (b) Vertical distribution of China

average daily mean total ozone simulated by control run and the sum of ozone contributed by domestic anthropogenic emissions, foreign anthropogenic emissions, natural sources which are calculated from sensitivity simulations.

Technical Corrections:

Throughout the manuscript there are many acronyms that are used but not defined (e.g. MOZART, NAQPMS, PKUCPL).

Modified as suggested. Thank you.

L91: The ozone itself doesn't differ, but the plumes and chemical regimes, which produce and destroy the ozone, does differ.

These sentences address the difference between 1) the transboundary ozone due to a particular region's emissions and 2) the ozone produced in the troposphere within the territory of that region from global precursor emissions.

L156-171: This paragraph mostly duplicates the information already in Table 1, and I do not feel that this redundancy is necessary.

Although we provided the emission inventories in Table 1, we felt that due to their importance to this study, it is better to also briefly describe these inventories in the main text to enhance readability and understanding.

L198-199: Figure 3 should be Figure 2

Fixed as suggested. Thank you.

L250: These numbers do not match those found in Table 3

Thanks for reminding us. We have modified the numbers both in the main text and in Fig. 3 and 4. The differences were due to a difference by mistake in the treatment of rounding.

L265-266: The authors claim that the biases are due to overestimated free tropospheric and stratospheric transport, but it's not clear to me how this conclusion was reached.

All stations shown in revised paper Line 285–287 are background stations with high altitude of more than 1500m. Ozone concentrations measured at these stations represent the background situation of the free troposphere, which is influenced by ozone transport from stratosphere.

The color scales in Figures 8a,b,d,e need to be consistent, as it requires extra effort to compare the Chinese Anthropogenic and Foreign Anthropogenic contributions. There is a risk that a casual reader would assume that the color scales in Figures 8a,b,d,e are the same, which would lead to incorrect conclusions.

Using the same color scales leads to loss of detailed information in the spatial variability of foreign anthropogenic $O_3$ and Ox, as shown in Fig. A5 below. Since this detailed information is of great interest in this study, we have elected to retain the original color scales and added a note in the caption that the color scales are different between (a, d) and (b, e).

[Figure]

Figure A5. Spatial distribution of springtime daily mean surface ozone over China contributed by (a) domestic and (b) foreign anthropogenic emissions. (c) Percentage contribution of foreign anthropogenic emissions to total anthropogenic ozone; areas with negative Chinese contributions (due to NOx titration) are marked in grey. (d–f) similar to (a–c) but for Ox (= O3 + NO2). The linear weighting adjustment is applied to derive all results.

L384: I don't feel that describing the air over the Sichuan Basin as "more isolated" is the correct description; rather the ozone chemistry of the region is controlled and dominated by domestic emissions and chemistry rather than foreign emissions.

Here we only consider surface ozone enhancement (in absolute terms, i.e., ppb) by foreign anthropogenic emissions, how much Chinese emissions contribute to ozone in this area is not relevant.

The relatively low ozone contribution from foreign emissions over Sichuan Basin compared to elsewhere may be caused by the "more isolated" terrain. Sichuan Basin is surrounded by high elevation mountains (new Fig. 3). The Qinghai-Tibet Plateau in the west and the Yunnan-Guizhou Plateau in the south block the airflows from South Asia and South-East Asia (new Fig. 10b and c). Qinling Mountains make the airflow from the north difficult to be transported to Sichuan Basin. (new Fig. 10e and f).

**Anonymous Referee #2**

The manuscript presents a modeling analysis and attributes ozone in China to anthropogenic emissions outside China. Basically, it represents a breakdown of background ozone in China to different foreign regions. Although this breakdown analysis has its value, my main concerns are (1) the analysis was limited to the seasonal mean ozone attribution rather than high ozone events, (2) the modeling was based on a single, non- recent, year (2008), and (3) the nonlinearity in source attribution seems to be large and needs to be assessed more carefully. These issues need to be addressed and corrected before this work can be accepted by ACP.

We thank the reviewer for thoughtful comments, which have been incorporated in the revised manuscript.

Major comments

1. The last sentence of the abstract, "Global emission reduction is critical for China's ozone mitigation", should be removed. The reported contribution of foreign emissions on ozone in China is essentially the background ozone. It has been well established (e.g. by several HTAP reports and references therein) that background ozone is substantial (20-50 ppbv) everywhere in the northern mid-latitude continents. For long-lived air pollutants such as ozone, essentially every country pollutes others and vice versa. To effectively mitigate ozone pollution in China, the key is to understand which source region drives the variability, especially of the high ozone days. I would be surprised if the foreign contribution is a primary factor for day-to-day changes of peak ozone over the majority of China. It appears that the paper only focuses on the seasonal mean contributions from foreign sources, thus the last sentence is a premature statement and may be interpreted misleadingly that domestic emissions control is not important.

Foreign contributed ozone can affect both the (seasonal) mean value of ozone in the receptor region as well as the peak ozone days. This study focuses on the mean impacts. Although the peak ozone days are an important aspect of ozone pollution, the mean value is of great interest. A large amount of existing ozone transport model studies are also focused on mean ozone (seasonal mean, seasonal MDA8, annual mean, etc.) (Verstraeten et al., 2016; Li et al., 2016b; Zhu et al., 2016). In fact, new epidemiological studies have suggested a strong impact of long-term mean ozone on human health, and that there is no threshold of ozone concentrations below which ozone exposure is not harmful (Bell et al., 2006; Yang et al., 2012; Peng et al., 2013; Di et al., 2017; Shindell et al., 2018).

Although a qualitative understanding has been reached that long-range transport of ozone is important, quantitative assessments are still scarce for transboundary impacts on China, as shown in the introduction section, especially compared to the large number of studies for the United States and some other countries. Although HTAP and earlier studies have worked on long-range transport impacts on Asia, the quantitative understanding for China is still poor due to this lack of China-focused studies. Also important, here we have used a comprehensive suite of near-surface and vertical profile measurements to constrain the model prior to source attribution calculations. Furthermore, as stated the introduction, we have analyzed not just the total impact of each particular foreign region but also separated the contribution of ozone produced within that source region and the contribution of ozone produced outside that source region (along the transport pathway). To our knowledge, this is the first time for China-focused transport studies.

We did not state that domestic emission control is not important. Instead, we argued, based on our detailed quantitative attribution calculations, that global emission control is important for Chinese ozone pollution mitigation. We have revised the statement to "In addition to domestic emission control, global emission reduction is critical for China's ozone mitigation".

References:

Bell, M. L., Peng, R. D., and Dominici, F.: The exposure-response curve for ozone and risk of mortality and the adequacy of current ozone regulations, Environ. Health Persp., 114, 532–536, https://doi.org/10.1289/ehp.8816, 2006

Di, Q., Wang, Y., Zanobetti, A., Wang, Y., Koutrakis, P., Choirat, C., Dominici, F., Schwartz, J. D.: Air Pollution and Mortality in the Medicare Population, New England Journal of Medicine, 376, 2513-2522, 10.1056/NEJMoa1702747, 2017

Peng, R. D., Samoli, E., Pham, L., Dominici, F., Touloumi, G., Ramsay, T., Burnett, R. T., Krewski, D., Le Tertre, A., Cohen, A., Atkinson, R. W., Anderson, H. R., Katsouyanni, K., and Samet, J. M.: Acute effects of ambient ozone on mortality in Europe and North America: results from the APHENA study, Air Qual. Atmos. Hlth., 6, 445–453, https://doi.org/10.1007/s11869-012- 0180-9, 2013.

Shindell, D., Faluvegi, G., Seltzer, K., Shindell, C.: Quantified, localized health benefits of accelerated carbon dioxide emissions reductions, Nature Climate Change, 8, 291–295, 10.1038/s41558-018-0108-y, 2018

Yang, C. X., Yang, H. B., Guo, S., Wang, Z. S., Xu, X. H., Duan, X. L., and Kan, H. D.: Alternative ozone metrics and daily mortality in Suzhou: The China Air Pollution and Health Effects Study (CAPES), Sci. Total Environ., 426, 83–89, https://doi.org/10.1016/j.scitotenv.2012.03.036, 2012

2. To follow up the previous comment, by reporting just seasonal mean contributions of foreign sources on ozone in China, I feel the paper does not add much new knowledge to the field, especially considering their analysis was based on a single year's simulation (see my next comment). The paper would be interesting if they had analyzed the foreign contribution to peak ozone events (during pollution episode) in addition to the mean ozone.

Please see our response regarding "peak ozone" above.

As added in our newly added Sect. 4.3 Line 482–491, here we show the domestic versus foreign contributions to modeled extreme ozone values in spring 2008 (defined as the average of the top 5% hourly ozone concentrations) (Fig. A6a–c). For comparison, we also adopt the results for mean ozone from Fig. 9 a–c (old Fig. 8a–c) and modify the color scale to make it consistent with Fig. A6a–c, as shown in Fig. A6d–f here. As expected, Chinese domestic contribution is larger for extreme ozone than for mean ozone; the negative values also disappear over North China Plain and Northeast China (comparing Fig. A6a and d). The absolute foreign contribution (in ppb) is also enhanced across China (comparing A6b and e). The percentage foreign contribution is within 10% over southern China, about 10–50% over the north, and above 70% over the west. Nevertheless, these results for extreme ozone should be interpreted with more caution, as the model cannot simulate the dates of extreme ozone very well (Fig. 4).

[Figure]

Figure A6. Spatial distribution of springtime extreme value (defined as the average of the

highest 5% hourly ozone concentrations) of surface ozone over China contributed by (a) domestic and (b) foreign anthropogenic emissions. (c) Percentage contribution of foreign anthropogenic emissions to total anthropogenic ozone; (d–f) similar to (a–c) but for daily mean surface ozone. Areas with negative Chinese contributions (due to NOx titration) are marked in grey. The linear weighting adjustment is applied to derive all results. Please note that the color scales are different between (a, d) and (b, e).

3. I have concerns about the choice of a single, non-recent, year (2008) used in the paper for the whole analysis. The exact magnitudes of ozone mixing ratio attributable to different sources depend on meteorology and emissions, both linked with the year of simulation. How would these ozone values change if another year is chosen to conduct the analysis? The authors stated that routine ozone measurements were scarce before 2013 (pg 6, line 203), so why not a simulation year after 2013? This would be more desirable to take advantage of more observational data for model evaluation. In particular, the increase of ozone pollution is a more recent concern in Chinese cities, after high PM events are on the decline.

We had thought about the choice of study year when conceiving the study, particularly whether to focus on a more recent year or not. At last, we decided to focus on 2008 for several reasons. First, for ozone transport model studies, it is important for model validation to have high quality observation data both near the surface and for the vertical profile that are representative of the regional ozone. The year of 2008 is when a comprehensive suite of near-surface and vertical profile measurements is available. And the observation data we used are high quality, well documented, and widely used in the literature. Although there are much more near-surface measurements from the Ministry of Environmental Protection (MEP) after 2013, there are few vertical profile measurements available in these more recent years. Also, the MEP measurements are almost all in the urban areas and cannot be used effectively to constrain the model, because our model resolution (0.5×0.667 degree) is not expected to capture the urban pollution chemistry well.

As in the newly added Sect. 4.3 (Line 492-516), previous studies have shown notable interannual variability in surface ozone over China driven by changes in precursor emissions and meteorology (Xu et al., 2008; Jin et al., 2015; Wang et al., 2017). To test how the interannual variability of meteorology and emissions would affect our source attribution findings, we have repeated all zero-out runs for spring 2012, the latest year when the GEOS-5 meteorological fields are available. Emissions for 2012 were adopted from the Community Emissions Data System (CEDS) inventory (Hoesly et al., 2018); 2012 is also the latest year the CEDS emissions for China are adjusted by the MEIC inventory. Table A1 shows the anthropogenic emissions in the two years. All zero-out simulation results in 2012 underwent the same linear weighting adjustment as for those in 2008. Figure A1d–f show the results for domestic versus foreign contributed ozone in spring 2012, as compared to the results for spring 2008 (adopted from Fig. 9a–c). In absolute terms, Chinese contributed ozone are similar between 2008 and 2012 (comparing Fig. A1a and d), reflecting the slight changes in domestic precursor emissions (Table A1). From 2008 to 2012, the absolute foreign contributed ozone increase along the southern boarder due to much enhanced emissions in South-East Asia and South Asia. The absolute foreign contributions decrease over the north and south, reflecting the net effect of changes in European and North American emissions (within 20% for both NOx and NMVOC), increased emissions in Rest of Asia, and changes in meteorology. In relative terms (Fig. A1c and f), the percentage foreign anthropogenic contributions to total anthropogenic ozone decrease from 2008 to 2012 over southern China. Nonetheless, in both years the percentage foreign contributions exceed 50% over western China and are 5–40% over southern China. Therefore our general finding that both foreign and domestic contributions to Chinese anthropogenic ozone are important holds true for these two years.

Further remarks: China is facing a sever ozone pollution problem, which has been getting worse in recent years. To tackle this problem domestic emission reductions (for both NOx and NMVOC) are of tremendous importance. Nonetheless, our results here show that foreign emission control is also necessary to ensure the success of ozone mitigation. This is particularly important during the time of fast economic growth and industrial development in nearby countries.

4. I am also concerned with the statement that over the polluted eastern China, "Chinese anthropogenic emissions lead to reductions (instead of enhancements) of surface ozone" (pg 10, line 374-375). The authors attributed this to the ozone titration effect by freshly emitted NO. The phenomena do occur in urban areas, but the GEOS-Chem simulation used in this study has a relatively coarse grid cell even for the nested-grid option (~50km x 50 km). This resolution would substantially smear out NOx emissions in a grid, leading to muted titration effect. My interpretation of that statement is that it suggests the nonlinearity in the zero-out simulations is strong (because it leads to negative ozone changes) and needs to be tested via different sensitivity runs and dealt with carefully. For example, the authors could try zeroing-out foreign anthropogenic emissions instead of Chinese anthropogenic emissions or try reducing Chinese emissions by a certain percentage rather than a complete zero-out, and then analyze if the different perturbation runs give consistent results over North China.

We agree that our model resolution cannot resolve the urban chemistry very well, which one of the reasons we had chosen not to focus our study on a more recent year and use the urban measurements from the MEP to validate the model. Nonetheless, at our model resolution, the spatial distribution of precursor emissions still show spatial contrast clearly, especially for NOx emissions, as shown in the plots below (adopted from Yan et al., 2016).

[Figure]

Figure A7. Total (anthropogenic and natural) emissions of NMVOCs and NOx over Asia, as represented in the nested model. Values outside the upper bound of color intervals are shown in black. Color intervals are nonlinear to better present the data range; an interval without labeling represents the mean of adjacent two intervals. Also depicted in each panel is the regional total. (Plots are adopted from Yan et al., 2016)

As suggested by the reviewer, we ran one more set of simulations by decreasing 20% anthropogenic emissions over each of the eight emission source regions (see the detailed information in Table A2). We also applied the linear weighting method to account for the non-linearity of ozone chemistry.

Figure A8a and d compares the Chinese anthropogenic contributed ozone calculated from 20%-perturbation and from zero-out simulations. Compared to the zero-out method, the 20% perturbation method leads to less Chinese contributed ozone, with negative values over more regions and smaller positive values over southern China. This result confirms our general finding that in spring 2008, the excessive domestic NOx emissions lead to relatively weak ozone production and/or strong ozone titration. Comparing to the zero-out method, the absolute foreign anthropogenic ozone obtained from 20%-perturbation

simulations are smaller by 2–3 ppb over the northern border of China (comparing Fig. A8b and e), whereas the percentage foreign contributions increase from 10–20% to 20–40% over southeastern China (comparing Fig A8c and f). Nonetheless, the spatial patterns are similar between the two methods for both the absolute and the relative foreign contributions.

We have added these results in the newly added Sect. 4.3 Line 462–481.

**Table A2.** Model simulations

| Full chemistry simulation | Description |
| --- | --- |
| CTL | Full-chemistry simulation with all emissions |
| $x_{20}$ANTH | Without 20% global anthropogenic emissions |
| $x_{20}$CH | Without 20% anthropogenic emissions of China |
| $x_{20}$JAKO | Without 20% anthropogenic emissions of Japan and Korea |
| $x_{20}$SEA | Without 20% anthropogenic emissions of South-East Asia |
| $x_{20}$SA | Without 20% anthropogenic emissions of South Asia |
| $x_{20}$ROA | Without 20% anthropogenic emissions of Rest of Asia |
| $x_{20}$EU | Without 20% anthropogenic emissions of Europe |
| $x_{20}$NA | Without 20% anthropogenic emissions of North America |
| $x_{20}$ROW | Without 20% anthropogenic emissions of Rest of World |

[Figure]

Figure A8. Spatial distribution of springtime daily mean surface ozone over China contributed by (a) domestic and (b) foreign anthropogenic emissions getting from 20%-perturbation method. (c) Percentage contribution of foreign anthropogenic emissions to total anthropogenic ozone; (d–f) similar to (a–c) but for zero-out method. Areas with negative Chinese contributions (due to NOx titration) are marked in grey. The linear weighting adjustment is applied to derive all results. Please note that the color scales are different between (a, d) and (b, e).

Minor Issues

Pg 5, line 190-195: The description of the weighting method to account for nonlinear chemistry is very vague, and I don't understand the scientific basis for this method. It should be expanded and explained in a way such that it is understandable to readers who have not read the original Li et al (2016) paper.

Ozone production is nonlinearly dependent on its precursors. Thus, the sum of natural ozone and anthropogenic ozone due to each emission source region calculated from zero-out simulations is not equal to ozone concentration calculated in the control run (CTL). Considering uncertainties induced by emission perturbation methods, we used a linear weighting method to adjust ozone concentration attributed to different sources, making the sum of natural ozone and anthropogenic ozone equal to amount of ozone simulated in CTL.

As clarified in the revised manuscript, here is an example to adjust Chinese contribution to ozone over China using the linear weighting approach. Equation A1 calculates the fractional Chinese contribution ($\alpha$) to the sum of ozone from individual anthropogenic source regions and from natural sources; the simulations involved are all full-chemistry runs (CTL, xCH, xEU, …, xANTH). Equation A2 applies the fractional contribution $\alpha$ to the total ozone in CTL to obtain the final adjusted Chinese contribution. These equations are used in the revised manuscript for better clarity; they are simply a transformed version of Eq. 1 in the original manuscript.

Similar adjustments were applied to other source regions, such that all results shown in our original manuscript are for "adjusted" ozone attribution through this linear weighting approach.

As shown in our revised manuscript Line 216–218, "A similar approach was used by Li et al. (2016a) to estimate the contribution of China to global radiative forcing, although in their study 20% (instead of 100%) of emissions over individual emission source regions are removed in the sensitivity simulations."

$$\alpha = \frac{\text{Con(CTL)} - \text{Con(xCH)}}{\sum_{i=1}^{8}[\text{Con(CTL)} - \text{Con(Ci)}] + \text{Con(xANTH)}} \tag{A1}$$

$$C_{CH} = \alpha \times \text{Con(CTL)} = \frac{\text{Con(CTL)} - \text{Con(xCH)}}{\sum_{i=1}^{8}[\text{Con(CTL)} - \text{Con(Ci)}] + \text{Con(xANTH)}} \times \text{Con(CTL)} \tag{A2}$$

Pg 3, line 108-109: this sentence is confusing. Do you mean there are 10 producing regions and 8 source regions? Why and how are they different?

The eight source regions represent emitter of precursor gases. The 10 producing regions include the troposphere of eight emitters, the troposphere of total oceanic regions, and the stratosphere. We have clarified these terms in the revised manuscript.

Language Issues: The paper has a few grammar errors and language issues, some examples listed below. I would suggest the authors proofread it more carefully during the revision stage.

We have checked grammar errors and language issues of the paper again and fixed them in the revised version. Thanks for reminding.

Pg 1, line 6: "mean bias at 10-15%" should be "mean bias of 10-15%"

Fixed as suggested. Thank you.

Pg 2, line 38: "at surface" should be "at the surface".

Fixed as suggested. Thank you.

Pg 10, line 356: "nature ozone" should be "natural ozone".

Fixed as suggested. Thank you.